# MASAM: Multimodal Adaptive Sharpness-Aware Minimization for Heterogeneous Data Fusion

**Zijie Chen**[1,2], **Kejing Yin**[1]*, **Wenfang Yao**[3], **William K. Cheung**[1], **Jing Qin**[4]

[1] Department of Computer Science, Hong Kong Baptist University
[2] Institute of Systems Medicine and Health Sciences, Hong Kong Baptist University
[3] Division of Artificial Intelligence, Lingnan University
[4] School of Nursing, The Hong Kong Polytechnic University

## Abstract

Multimodal learning requires integrating heterogeneous modalities, such as structured records, visual imagery, and temporal signals. It has been revealed that this heterogeneity causes modality encoders to converge at different rates, making the multimodal learning imbalanced. We empirically observe that such an imbalance is related to the sharpness of the solution. Modality encoders that converge faster could be dragged into sharp regions due to inter-modal interference, degrading the generalization capability of unimodal features learned. Sharpness-Aware Minimization is effective in improving generalization via finding solutions in flat regions. However, its application in multimodal scenarios is challenging: 1) SAM overemphasizes the dominant modality, inducing misaligned perturbations in weaker modalities, and 2) the perturbation gradient calculation is affected by interference from other modalities. To address these issues, we propose Multimodal Adaptive Sharpness-Aware Minimization (MASAM), which optimizes different modalities based on their dominance. We design an Adaptive Perturbation Score (APS) using convergence speed and gradient alignment to identify dominant modalities for SAM application. Our Modality-Decoupled Perturbation Scaling (MDPS) then reduces inter-modal interference during optimization, better aligning each modality with shared information. Extensive empirical evaluations on five multimodal datasets and six downstream tasks demonstrate that MASAM consistently attains flatter solutions, achieves balanced multimodal learning, and subsequently surpasses state-of-the-art methods across diverse datasets and tasks. Code is available at https://github.com/Orange2107/MASAM-Multimodal-Adaptive-SAM.

## 1 Introduction

Multimodal learning leverages complementary information from heterogeneous data sources and has achieved remarkable progress in domains such as audio–video understanding (Huang et al., 2023) and clinical decision support (Stahlschmidt et al., 2022). Despite these successes, prior studies have highlighted the persistent issue of modality imbalance, where faster-converging modalities dominate training and weaker modalities are under-optimized, often leading to suboptimal fusion or even inferior performance compared to unimodal models (Wang et al., 2020; Peng et al., 2022; Wei & Hu, 2024). A range of strategies, gradient modulation (Li et al., 2023), iterative unimodal optimization (Zhang et al., 2024), and representation regularization (Fan et al., 2023), have been developed to alleviate this problem, but these approaches primarily modulate gradient magnitudes and overlook deeper geometric factors, such as the sharpness of the loss landscape and the solutions.

Such geometric factors have a close linkage to the generalization capability and also modality balance in the multimodal settings: flatter minima typically correspond to more robust and stable solutions (Hochreiter & Schmidhuber, 1997; Keskar et al., 2017; Foret et al., 2021). In multimodal settings, heterogeneous modalities exhibit inherently divergent convergence dynamics. Even when a

---

*Correspondence author: Kejing Yin (cskjyin@comp.hkbu.edu.hk).

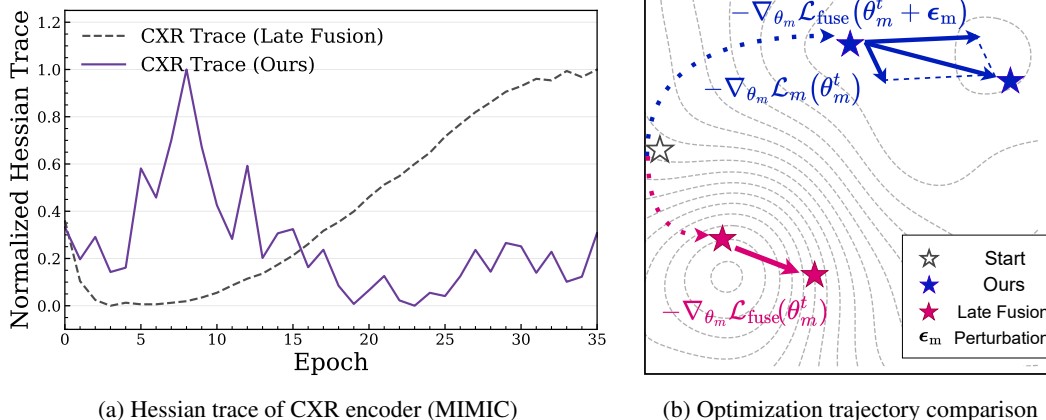

(a) Hessian trace of CXR encoder (MIMIC)  (b) Optimization trajectory comparison

Figure 1: (a) The Hessian trace analysis for CXR modality in the MIMIC dataset, where a lower normalized Hessian trace represents a flatter minima. It suggests that the CXR encoder in the naive late fusion model converges to a sharp region, whereas that in our MASAM converges to a much flatter region. (b) An illustration of the optimization trajectory of naive late fusion and MASAM.

modality progresses towards a flatter solution, joint optimization can continually disrupt its trajectory through cross-modal interference, preventing the model from reaching a jointly flat optimum and thereby making multimodal learning imbalanced and limiting its generalization capacity. We empirically observe this phenomenon in real-world datasets. Fig. 1a plots the Hessian trace (a proxy for sharpness) of modality-specific encoders for the MIMIC dataset (EHR + CXR), where the CXR encoder initially converges towards a flatter region (smaller Hessian Trace) but later becomes sharper as the unstable optimization of the EHR encoder disrupts its trajectory. Fig. 1b provides a conceptual illustration, showing that cross-modal interference under Late Fusion diverts the optimization trajectory into sharper regions.

Sharpness-Aware Minimization (SAM) has demonstrated consistent improvements in generalization for single-modality deep learning by steering optimization towards flat regions of the loss landscape (Foret et al., 2021). This benefit stems from the fact that flat minima are associated with robustness to parameter perturbations and data variability (Hochreiter & Schmidhuber, 1997; Keskar et al., 2017), making SAM a widely adopted technique for enhancing model reliability in diverse domains. We conjecture that similar ideas could be used to mitigate the modality imbalance issue in multimodal learning. However, directly applying SAM to multimodal settings introduces fundamental challenges that undermine its effectiveness: **(1) Amplified modality imbalance**: Different modalities exhibit heterogeneous convergence speeds and gradient magnitudes. SAM's uniform sharpness minimization thus disproportionately benefits dominant modalities with faster convergence, while under-optimizing slower yet informative

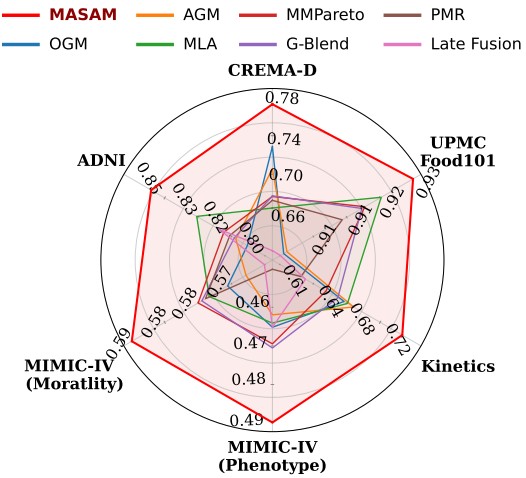

Figure 2: The proposed MASAM model achieves the best performance across all datasets and tasks.

ones (Fan et al., 2024). **(2) Modality-agnostic perturbation**: Beyond imbalance in magnitude, modalities also possess distinct loss landscape geometries. The perturbation applied by SAM is agnostic to differences in loss landscape geometries, forcing encoders of different modalities to adapt to a shared trajectory rather than their own flat regions. This one-size-fits-all approach leads to undermined multimodal generalization.

To address these challenges, we propose **Multimodal Adaptive SAM (MASAM)**, an optimization framework tailored for multimodal learning. To mitigate amplified modality imbalance, MASAM introduces an Adaptive Perturbation Score (APS) to quantify the dominance of each modality through its convergence speed and gradient alignment with the fusion objective. Perturbations are then selectively applied to dominant modalities, stabilizing their convergence in flatter regions while preventing weaker modalities from being affected by misaligned perturbations from the dominant modality. To address modality-agnostic perturbation, MASAM develops a Modality-Decoupled Perturbation Scaling (MDPS) that adjusts perturbation strength according to cross-modal gradient alignment. This enables each modality to explore flatter minima along directions that align with shared information while remaining robust to heterogeneous loss landscapes. Together, these mechanisms make SAM modality-aware, allowing multimodal models to retain the benefits of flatness-driven generalization without suffering from imbalance or incompatible perturbations. Fig. 2 summarizes the quantitative evaluation on six diverse multimodal datasets, demonstrating the effectiveness of MASAM.

Our main contributions are summarized as follows:

- We identify two fundamental limitations: amplified modality imbalance and modality-agnostic perturbation, which hinder the direct application of SAM to multimodal learning.
- We propose MASAM, a multimodal optimization framework that explicitly tackles modality imbalance and heterogeneous loss geometries through Adaptive Perturbation Score (APS) and Modality-Decoupled Perturbation Scaling (MDPS).
- We conduct extensive experiments across diverse domains, from widely used multimodal benchmarks (CREMA-D, Kinetics-Sounds, UPMC-Food101) to noisy clinical datasets (MIMIC-IV, ADNI), and extend MASAM to tri-modal learning on UR-FUNNY. Results show that MASAM significantly outperforms existing baselines, consistently addresses modality imbalance issues by achieving flatter optima, and obtains better robustness against noisy labels.

## 2  PRELIMINARIES AND RELATED WORK

**Preliminaries on Sharpness-Aware Minimization**  Sharpness-Aware Minimization (SAM) (Foret et al., 2021) seeks parameters in flat regions in the loss landscape. Formally, SAM minimizes both the loss value and the sharpness of the loss surface, where sharpness is defined as the maximum increase in loss under a small perturbation $\epsilon$ within an $l_2$-ball of radius $\rho$ of the parameters $\boldsymbol{\theta}$:

$$\min_{\boldsymbol{\theta}} \left[ \max_{\|\boldsymbol{\epsilon}\|_2 \leq \rho} \mathcal{L}(\boldsymbol{\theta} + \boldsymbol{\epsilon}) - \mathcal{L}(\boldsymbol{\theta}) \right] + \mathcal{L}(\boldsymbol{\theta}). \tag{1}$$

By applying a first-order Taylor expansion, the inner maximization reduces to perturbing parameters in the gradient direction:

$$\min_{\boldsymbol{\theta}} \mathcal{L}(\boldsymbol{\theta} + \hat{\boldsymbol{\epsilon}}), \text{ where } \hat{\boldsymbol{\epsilon}} \triangleq \rho \cdot \frac{\nabla \mathcal{L}(\boldsymbol{\theta})}{\|\nabla \mathcal{L}(\boldsymbol{\theta})\|}. \tag{2}$$

Intuitively, this procedure penalizes sharp directions in the loss surface, steering training towards flatter minima that enhance generalization and robustness (Andriushchenko et al., 2023). Several variants of SAM refine the perturbation or sharpness estimation (Wang et al., 2023; Wu et al., 2024a; Li et al., 2024), but all are developed for unimodal tasks, leaving their applicability to multimodal optimization unexplored.

**Related Work on Balancing Multimodal Learning**  Recent efforts have been made to mitigate the modality competition issue, where the learning is dominated by stronger modalities. Existing works mostly rely on gradient modulation, such as G-Blend (Wang et al., 2020), OGM (Peng et al., 2022), and AGM (Li et al., 2023). They tackle the modality imbalance by dynamically rescaling gradients, but do not take the geometry of the loss surface into consideration, potentially leading to poor generalization. For more discussions of related work, see Appendix B.

**Why SAM Fails in Multimodal Learning?**  While SAM and its variants have demonstrated consistent improvements in unimodal learning, directly applying SAM to multimodal learning is problematic since the gradients of the fusion objective are coupled across modalities. When

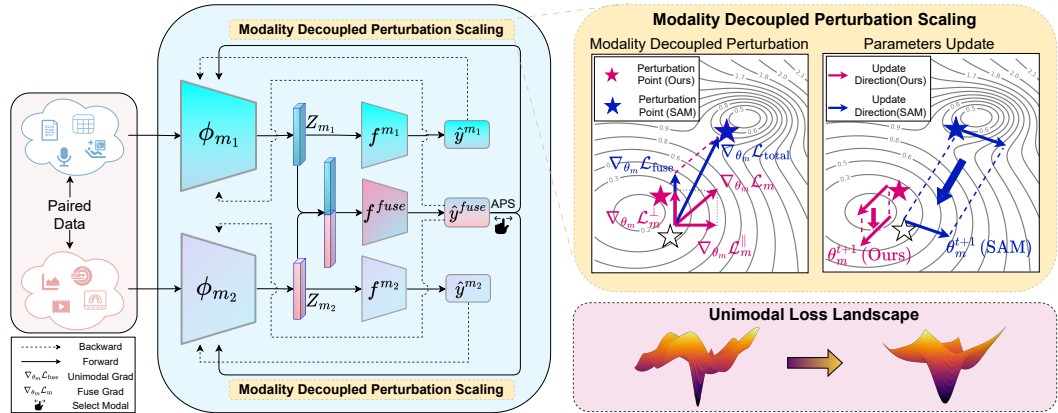

Figure 3: Overview of MASAM. We propose to adaptively apply perturbations to specific modalities, guiding each modality towards its own flat minimum to address modality imbalance. First, we introduce the Adaptive Perturbation Score (APS) to identify strong modalities in training. Then, Modality-Decoupled Perturbation Scaling (MDPS) aligns gradients between unimodal and fusion objectives, mitigating cross-modal interference under SAM perturbations.

one modality dominates the prediction, weaker modalities receive distorted gradients, and SAM perturbations derived from them fail to ensure modality-specific flatness. We formally summarize this observation as follows.

**Observation 1** (Difficulties of applying SAM to multimodal learning). *Consider multimodal learning with at least two modalities, trained under a joint late fusion loss $\mathcal{L}_{\text{fuse}}$. If one modality is dominant (i.e., contributes disproportionately to the fused prediction), then:*

1. *The gradient $\nabla_{\boldsymbol{\theta}_m}\mathcal{L}_{\text{fuse}}$ for each weaker modality $m$ is biased towards the dominant modality, rather than aligned with its own unimodal objective.*

2. *The SAM perturbation direction for modality $m$*

$$\boldsymbol{\epsilon}_m = \rho \cdot \frac{\nabla_{\boldsymbol{\theta}_m}\mathcal{L}_{\text{fuse}}}{\|\nabla_{\boldsymbol{\theta}_m}\mathcal{L}_{\text{fuse}}\|_2} \tag{3}$$

*is therefore also biased, preventing weaker modalities from converging to their flat minima.*

*As a result, applying SAM directly to multimodal training can exacerbate modality imbalance and fail to achieve modality-specific flatness.*

We provide a detailed derivation in Appendix C. This observation shows that SAM's perturbation mechanism, while effective in unimodal training, becomes counterproductive in multimodal settings, motivating the modality-adaptive variant we propose in this paper.

## 3 MASAM: MULTIMODAL ADAPTIVE SHARPNESS-AWARE MINIMIZATION

### 3.1 THE MASAM FRAMEWORK

Prior analysis demonstrates that in multimodal training, the dominant modality often steers the joint optimization process, resulting in insufficient exploration and suboptimal learning for weaker modalities. Moreover, the perturbation direction in SAM tends to align with the stronger modality's gradient, further impeding the weaker modality's ability to converge to flatter minima. To mitigate these challenges, we propose a method that dynamically selects which modality to perturb and enforces gradient alignment across modalities. The overall framework is illustrated in Fig. 3.

**Multimodal fusion** Without loss of generality, we assume two modalities $m_1$ and $m_2$. For each data sample indexed by $i$, we consider a paired multimodal instance with the downstream task label denoted as $(x_i^{m_1}, x_i^{m_2}, y_i)$. The data modalities $x_i^{m_1}$ and $x_i^{m_2}$ are encoded using modality-specific encoders, denoted as $\phi_{m_1}$ and $\phi_{m_2}$ and parameterized by $\boldsymbol{\theta}_{m_1}$ and $\boldsymbol{\theta}_{m_2}$, respectively. Then, they are combined for downstream tasks.

$$\hat{y}_i^{\text{fuse}} = f^{\text{fuse}}(\mathbf{z}_i^{\text{fuse}}), \qquad \mathbf{z}_i^{\text{fuse}} = \text{Fusion}(\mathbf{z}_i^{m_1}, \mathbf{z}_i^{m_2}; \boldsymbol{\theta}_{\text{fuse}}) = \mathbf{W}\left[\mathbf{z}_i^{m_1}||\mathbf{z}_i^{m_2}\right] + \mathbf{b}, \tag{4}$$

$$\mathbf{z}_i^{m_1} = \phi_{m_1}(x_i^{m_1}; \boldsymbol{\theta}_{m_1}), \quad \mathbf{z}_i^{m_2} = \phi_{m_2}(x_i^{m_2}; \boldsymbol{\theta}_{m_2}), \tag{5}$$

where $||$ denotes concatenation, $f^{\text{fuse}}$ is the task-specific classification head, and $\hat{y}_i^{\text{fuse}}$ is the final prediction. We adopt a unified cross-entropy objective, denoted as CE. The fusion loss $\mathcal{L}_{\text{fuse}}$ and the auxiliary unimodal objective $\mathcal{L}_m$ of modality $m$ are defined as:

$$\mathcal{L}_{\text{fuse}} = \frac{1}{N} \sum_{i=1}^{N} \text{CE}(\mathbf{y}_i, \hat{\mathbf{y}}_i^{\text{fuse}}), \quad \mathcal{L}_m = \frac{1}{N} \sum_{i=1}^{N} \text{CE}(\mathbf{y}_i, \hat{\mathbf{y}}_i^m), \quad m \in \{m_1, m_2\}, \tag{6}$$

where $\hat{\mathbf{y}}_i^m = f^m(\mathbf{z}_i^m)$ is the prediction from modality $m$ using its unimodal feature representation. The final training objective combines the fusion loss and unimodal auxiliary losses:

$$\mathcal{L}_{\text{total}} = \mathcal{L}_{\text{fuse}} + \lambda_{m_1} \mathcal{L}_{m_1} + \lambda_{m_2} \mathcal{L}_{m_2}, \tag{7}$$

where $\lambda_{m_1}$ and $\lambda_{m_2}$ are hyperparameters that control the strengths of the unimodal objectives.

**Adaptive perturbation score (APS)** As noted in prior works (Wang et al., 2020; Li et al., 2023), different modalities exhibit distinct convergence behaviors and make uneven contributions to the joint optimization process during training. A modality with a faster decrease of the uni-modal loss $\mathcal{L}_m$ typically indicates that its information is being efficiently and continuously learned, thus suggesting a stronger contribution. To capture this phenomenon, we introduce a measure for modality learning speed based on moving averages:

$$\text{Decay}_m^{(t)} = \max\left(0, \mathcal{L}_m^{(t-1)} - \text{MA}_m^{(t)}\right), \ m \in \{\text{m}_1, \text{m}_2\}, \tag{8}$$

where $\text{MA}_m^{(t)} = \beta \cdot \text{MA}_m^{(t-1)} + (1-\beta) \cdot \mathcal{L}_m^{(t)}$ is the moving average of the loss values during training, $t$ denotes the number of training steps. $\text{MA}_m^{(t)}$ provides a stable estimate of the long-term loss trend, while $\text{Decay}_m^{(t)}$ reflects the short-term reduction relative to this trend. A higher decay value indicates faster learning progress, implying that the modality is contributing more actively at the current step.

In addition to learning speed, we also consider gradient alignment as an indicator of modality strength. Our earlier analysis suggests that strong modalities, which convey richer shared information, tend to dominate the optimization trajectory of the fusion objective. This manifests as a high alignment between the gradients of the uni-modal loss and that of the fusion loss, since the latter is largely steered by the stronger modality. Conversely, weaker modalities may have misaligned or even conflicting gradient directions with respect to the fusion gradient, resulting in limited influence during joint learning. This perspective is also observed in Fan et al. (2023) and Guo et al. (2024). We therefore define a gradient-based modality dominance metric as:

$$\gamma_m^{(t)} = \frac{\langle \nabla_{\boldsymbol{\theta}_m} \mathcal{L}_{\text{fuse}}, \nabla_{\boldsymbol{\theta}_m} \mathcal{L}_m \rangle}{\|\nabla_{\boldsymbol{\theta}_m} \mathcal{L}_{\text{fuse}}\|_2 \cdot \|\nabla_{\boldsymbol{\theta}_m} \mathcal{L}_m\|_2}, \quad m \in \{\text{m}_1, \text{m}_2\}, \tag{9}$$

where $\nabla_{\boldsymbol{\theta}_m} \mathcal{L}_m$ and $\nabla_{\boldsymbol{\theta}_m} \mathcal{L}_{\text{fuse}}$ denote the gradients from the unimodal objective and the fusion objective, respectively. This unimodal loss not only promotes the acquisition of modality-specific information but also serves as a proxy to assess the modality's standalone learning dynamics. By combining both perspectives, we define our Adaptive Perturbation Score (APS) as:

$$\text{APS}_m^{(t)} = \alpha \cdot \text{Decay}_m^{(t)} + (1-\alpha) \cdot \gamma_m^{(t)}, \quad m \in \{\text{m}_1, \text{m}_2\}, \tag{10}$$

where $\alpha \in [0, 1]$ is a tunable hyperparameter that balances learning speed and gradient consistency in determining the dominance of each modality. To mitigate the risk of strong modalities being pushed away from flatter minima due to the continued updates driven by weaker modalities, we first identify the strong modality as the one with a higher APS in each training step. Then, we impose modality-specific SAM-based regularization over the strong modality.

**Modality-decoupled perturbation scaling (MDPS)** In joint learning, the final prediction is derived from the output of the fusion module, which primarily captures modality-shared information that is directly relevant to the downstream task. Consequently, the gradient of the fusion objective, $\nabla_{\boldsymbol{\theta}_m} \mathcal{L}_{\text{fuse}}$, can be interpreted as the direction that guides the learning of shared representations. Motivated by this, we apply perturbations along the fusion gradient direction to the dominant modality identified by APS. However, as discussed in Observation 1 and Appendix C, the gradient $\nabla_{\boldsymbol{\theta}_m} \mathcal{L}_{\text{fuse}}$ has a coupled

---

**Algorithm 1:** MASAM: Multimodal Adaptive Sharpness-Aware Minimization

---

**Input:** Training dataset $\mathcal{D}$; Perturbation radius $\rho$; weights $\{\lambda_m\}$; momentum $\mu$; score weight $\alpha$.

**Output:** Modality encoder parameters $\{\theta^m\}_{m=1}^M$ and other model parameters $\theta^{\text{other}}$.

1 **for** *each iteration with mini-batch $S \subset \mathcal{D}$* **do**
2 $\quad$ Compute $\mathcal{L}_{\text{fuse}}(S)$ and $\{\mathcal{L}_m(S)\}_{m=1}^M$;
3 $\quad$ Update $\theta^{\text{other}}$ by gradient descent;
4 $\quad$ **for** *each modality $m \in \{1, \dots, M\}$* **do**
5 $\quad\quad$ Compute modality-specific APS score $\text{APS}_m$ using Eqs. (8) to (10);
6 $\quad$ Select dominant modality by $m^\star = \arg\max_m \text{APS}_m$;
7 $\quad$ Obtain the perturbed parameters $\theta_{m^\star} + \epsilon_{m^\star}$ using Eq. (11);
8 $\quad$ Update dominant modality encoder parameters using Eq. (12);
9 $\quad$ **for** *each non-dominant modality $m \in \{1, \dots, M\} \setminus \{m^\star\}$* **do**
10 $\quad\quad$ Update modality encoder parameter $\theta_m$ using Eq. (13);

---

effect from different modalities; therefore, directly applying the SAM perturbation may mislead weaker modalities during optimization. To address this issue, we propose a method that dynamically adjusts the perturbation magnitude based on the degree of alignment between unimodal and fusion gradients, yielding

$$\epsilon_{\text{m}} = \rho \cdot \gamma_m \cdot \frac{\nabla_{\theta_{\text{m}}}\mathcal{L}_{\text{fuse}}}{\|\nabla_{\theta_{\text{m}}}\mathcal{L}_{\text{fuse}}\|_2} = \rho \cdot \frac{\langle \nabla_{\theta_m}\mathcal{L}_{\text{fuse}}, \nabla_{\theta_m}\mathcal{L}_m\rangle}{\|\nabla_{\theta_m}\mathcal{L}_{\text{fuse}}\|_2 \cdot \|\nabla_{\theta_m}\mathcal{L}_m\|_2} \cdot \frac{\nabla_{\theta_{\text{m}}}\mathcal{L}_{\text{fuse}}}{\|\nabla_{\theta_{\text{m}}}\mathcal{L}_{\text{fuse}}\|_2}, \quad (11)$$

where $\epsilon_{\text{m}}$ is the perturbation applied to $\theta_m$, $\rho$ is a scalar that controls the step size of the perturbation (consistent with the definition in Eq. (3)), and $\gamma_m$ is the cosine similarity between the unimodal and fusion gradients, as defined in Eq. (9). This scaling can also be interpreted as a projection of the gradient of unimodal loss onto the gradient direction of the fusion loss, thus achieving modality-decoupled perturbation.

**Parameter Update** We divide all model parameters into three disjoint subsets, where $\{\theta_{m^\star}\}$ and $\{\theta_m | m \neq m^\star\}$ denote the parameters of the dominating and non-dominating modalities, respectively, and $\theta^{\text{other}}$ denotes all other model parameters. In each iteration, we first update $\theta^{\text{other}}$ using a base optimizer, e.g., SGD or Adam. Then, we identify the dominant modality using the APS score by $m^\star = \arg\max_m \text{APS}_m$, MASAM then employs MDPS as defined in Eq. (11) to perform sharpness-aware optimization that eliminates interference from other modalities. The update direction is given by the combination of the perturbed fusion gradient and the unimodal loss gradient:

$$\theta_m^{t+1} = \theta_m^t - \eta_t \left[\nabla_{\theta_m}\mathcal{L}_{\text{fuse}}\left(\theta_m^t + \rho_t \gamma_m^t \frac{\nabla_{\theta_{\text{m}}}\mathcal{L}_{\text{fuse}}^t}{\|\nabla_{\theta_{\text{m}}}\mathcal{L}_{\text{fuse}}^t\|_2}\right) + \nabla_{\theta_m}\mathcal{L}_m(\theta_m^t)\right], \ m \in \{m^\star\}. \quad (12)$$

For each non-dominant modality, MASAM updates the parameters using the gradients of both the fusion and unimodal objectives computed at the current parameters:

$$\theta_m^{t+1} = \theta_m^t - \eta_t \left[\nabla_{\theta_m}\mathcal{L}_{\text{fuse}}\left(\theta_m^t\right) + \nabla_{\theta_m}\mathcal{L}_m\left(\theta_m^t\right)\right], \quad m \neq m^\star. \quad (13)$$

We summarize the learning algorithm for an arbitrary number of modalities in Algorithm 1.

### 3.2 THEORETICAL ANALYSIS

We analyze the convergence of MASAM under standard smoothness and step size conditions, building on the inexact gradient descent (IGD) framework of Khanh et al. (2024).

**Theorem 1** (Convergence of MASAM). *For the modality-specific MASAM update as in Eq. (12): suppose the fusion loss $\mathcal{L}_{\text{fuse}}$ has $L_{\text{fuse}}$-Lipschitz continuous gradients with $L_{\text{fuse}} > 0$. Assume the learning rate sequence $\{\eta_t\}$ and the perturbation radius sequence $\{\rho_t\}$ satisfy*

$$\text{(i) } \sum_{t=1}^{\infty} \eta_t = \infty, \quad \text{(ii) } \eta_t \downarrow 0, \quad \text{(iii) } \sum_{t=1}^{\infty} \eta_t \rho_t < \infty, \quad \text{(iv) } \limsup_{t\to\infty} \rho_t < \frac{2}{L_{\text{fuse}}}. \quad (14)$$

*Then $\{\theta_m^t\}$ converges to a stationary point of the multimodal objective Eq. (7).*

The complete proof of this result is presented in Appendix D. The conditions in Theorem 1 are practical in implementation, given that the fusion loss is cross-entropy loss and the radius sequence $\{\rho_t\}$ can approximate a constant (see Khanh et al., 2024, Remark 3.6).

Table 1: Unified comparison across datasets. Columns report the primary evaluation metric used for each dataset. Underlined values indicate the strongest baseline, while bold highlights our method. Relative Gain is computed against the strongest baseline in each column. All results are averaged over four runs with different random seeds.

| Method | Phenotype AUPRC | Mortality AUPRC | CREMA-D Accuracy | KS Accuracy | UPMC-Food101 Accuracy | ADNI mAP | UR-FUNNY Accuracy |
|---|---|---|---|---|---|---|---|
| Late Fusion | $0.475_{\pm0.0020}$ | $0.567_{\pm0.0121}$ | $0.660_{\pm0.0209}$ | $0.636_{\pm0.0117}$ | $0.907_{\pm0.0015}$ | $0.826_{\pm0.0069}$ | $0.620_{\pm0.0164}$ |
| G-Blend (Wang et al., 2020) | $0.480_{\pm0.0029}$ | $0.584_{\pm0.0026}$ | $0.717_{\pm0.0134}$ | $0.670_{\pm0.0091}$ | $0.924_{\pm0.0021}$ | $0.818_{\pm0.0120}$ | N.A.[§] |
| OGM (Peng et al., 2022) | $0.475_{\pm0.0012}$ | $0.577_{\pm0.0092}$ | $0.769_{\pm0.0040}$ | $0.677_{\pm0.0092}$ | $0.907_{\pm0.0009}$ | $0.812_{\pm0.0200}$ | $\underline{0.632}_{\pm0.0098}$ |
| AGM (Li et al., 2023) | $0.472_{\pm0.0012}$ | $0.572_{\pm0.0042}$ | $0.746_{\pm0.0106}$ | $0.688_{\pm0.0027}$ | $0.908_{\pm0.0010}$ | $0.817_{\pm0.0160}$ | $0.630_{\pm0.0091}$ |
| MLA (Zhang et al., 2024) | $0.474_{\pm0.0006}$ | $0.582_{\pm0.0006}$ | $0.705_{\pm0.0033}$ | $0.681_{\pm0.0132}$ | $\underline{0.928}_{\pm0.0011}$ | $0.836_{\pm0.0095}$ | $0.624_{\pm0.0129}$ |
| MMPareto (Wei & Hu, 2024) | $0.479_{\pm0.0027}$ | $\underline{0.585}_{\pm0.0089}$ | $0.717_{\pm0.0106}$ | $0.658_{\pm0.0067}$ | $0.925_{\pm0.0006}$ | $0.823_{\pm0.0110}$ | N.A.[§] |
| PMR (Fan et al., 2023) | N.A.[†] | $0.572_{\pm0.0046}$ | $0.713_{\pm0.0309}$ | $0.631_{\pm0.0142}$ | $0.920_{\pm0.0015}$ | $0.820_{\pm0.0082}$ | N.A.[§] |
| InfoREG (Huang et al., 2025) | $\underline{0.481}_{\pm0.0017}$ | $0.581_{\pm0.0105}$ | $0.723_{\pm0.0054}$ | $0.671_{\pm0.0103}$ | $0.925_{\pm0.0008}$ | $0.820_{\pm0.0135}$ | $0.625_{\pm0.0084}$ |
| AUG (Jiang et al., 2025) | $0.472_{\pm0.0031}$ | $0.579_{\pm0.0120}$ | $\underline{0.770}_{\pm0.0152}$ | $\underline{0.689}_{\pm0.0083}$ | $0.923_{\pm0.0015}$ | $\underline{0.847}_{\pm0.0068}$ | N.A.[§] |
| DrFuse (Yao et al., 2024) | $0.479_{\pm0.0012}$ | $0.576_{\pm0.0034}$ | N.A.[‡] | N.A.[‡] | N.A.[‡] | $0.836_{\pm0.0044}$ | N.A.[§] |
| MedFuse (Hayat et al., 2022) | $0.470_{\pm0.0017}$ | $0.549_{\pm0.0018}$ | N.A.[‡] | N.A.[‡] | N.A.[‡] | $0.828_{\pm0.0115}$ | N.A.[§] |
| **MASAM (ours)** | $\mathbf{0.498}_{\pm0.0010}$ | $\mathbf{0.603}_{\pm0.0086}$ | $\mathbf{0.814}_{\pm0.0046}$ | $\mathbf{0.740}_{\pm0.0084}$ | $\mathbf{0.935}_{\pm0.0011}$ | $\mathbf{0.857}_{\pm0.0042}$ | $\mathbf{0.644}_{\pm0.0091}$ |
| *Relative Gain (%)* | *+3.53* | *+3.08* | *+5.71* | *+7.40* | *+0.75* | *+1.18* | *+1.90* |

[†] PMR relies on a contrastive learning module, which cannot be applied to the multi-label phenotype task.

[‡] DrFuse and MedFuse are tailored for clinical data; thus, we exclude them for datasets and tasks in other domains.

[§] These baselines are not designed for the tri-modality setting.

## 4 EXPERIMENTS AND RESULTS

We evaluate the proposed method on diverse multimodal datasets: two audio–video datasets, **CREMA-D** (Cao et al., 2014) and **Kinetics-Sounds (KS)** (Arandjelovic & Zisserman, 2017), one image–text dataset, **UPMC-Food101** (Wang et al., 2015), two clinical multimodal datasets, **MIMIC** (Johnson et al., 2023; 2019) and **ADNI** (Jack Jr et al., 2008), and a tri-modality dataset UR-FUNNY (Hasan et al., 2019). On the MIMIC dataset, we perform two clinical tasks, the **Phenotype Classification** and the **Mortality Prediction**. For comparison, we include several representative methods for addressing modality imbalance, including **InfoReg** (Huang et al., 2025), **AUG** (Jiang et al., 2025), **OGM** (Peng et al., 2022), **AGM** (Li et al., 2023), **MMPareto** (Wei & Hu, 2024), **PMR** (Fan et al., 2023), **MLA** (Zhang et al., 2024), and **G-Blend** (Wang et al., 2020), as baseline models. For clinical datasets, we additionally include **DrFuse** (Yao et al., 2024) and **MedFuse** (Hayat et al., 2022), which are tailored for clinical multimodal learning. Due to page limits, we report AUPRC for the two MIMIC tasks, mAP for ADNI, and accuracy for CREMA-D, KS, UPMC-Food101, and UR-FUNNY, with additional metrics in Appendix E.3.

We follow pipelines from existing works to preprocess the datasets, and split them into training, validation, and test sets with a 7:1:2 ratio. We use identical encoders across all models within each dataset for fair comparisons and determine the hyperparameters via grid search. All experiments are performed using a server with four NVIDIA RTX 4090 GPUs. The details of data preprocessing and hyperparameter settings can be found in Appendices A.3 and A.4.

### 4.1 OVERALL PERFORMANCE ACROSS DATASETS

**MASAM consistently achieves significant improvements across all datasets.** The overall experimental results are summarized in Table 1 and Fig. 2. On the KS dataset, MASAM attains 0.74 accuracy, providing a substantial **7.4%** improvement over the strongest baseline (AUG). Similar performance gain is also observed on CREMA-D, where MASAM reaches 0.814 accuracy with a **5.71%** relative gain. On clinical datasets, despite their high noise levels and missing rates (statistics in Appendix A.4), MASAM achieves reliable improvements over the best baselines: **3.53%** for phenotype and **3.08%** for mortality on MIMIC, and **1.18%** higher mAP on ADNI. We provide detailed MIMIC performance analysis across individual disease phenotypes in Appendix E.1. On UPMC-Food101, MASAM achieves a **0.75%** accuracy gain; although the improvement is modest, we perform a statistical test that confirms the significance of the improvement ($p = 0.0046 < 0.005$). On UR-FUNNY, MASAM achieves 0.641 accuracy, yielding a **1.42%** improvement over the strongest baseline. Overall, the promising results show that MASAM consistently outperforms existing methods across diverse and highly heterogeneous multimodal datasets.

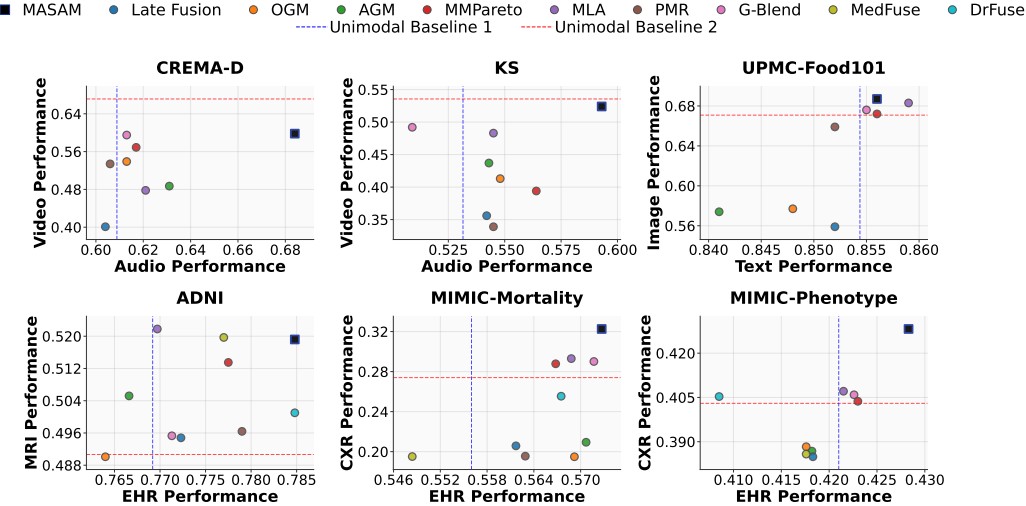

Figure 4: Unimodal performance across different datasets and modalities. *Unimodal Baseline* refers to a model trained and evaluated using only a single modality. Dots closer to the upper-right corner indicate stronger unimodal performance.

## 4.2 MODALITY BALANCE AND LOSS LANDSCAPE FLATNESS

**MASAM consistently improves unimodal performance across all datasets, demonstrating balanced multimodal learning.** We further evaluate unimodal performance by freezing the encoders obtained from multimodal training and training a classifier head on top. The results are summarized in Fig. 4, where dots closer to the upper-right corner indicate overall stronger single-modality performance. On CREMA-D and KS, MASAM surpasses all baselines, exceeding the unimodal Audio baseline and approaching the unimodal Video baseline. On the clinical dataset MIMIC, which is characterized by substantial noise and missingness, most baselines fail to obtain better unimodal performance than the models trained using unimodal data, indicating severe modality imbalance during learning. In contrast, MASAM demonstrates a significant advantage: it achieves consistent superiority over all multimodal baselines, and also significantly exceeds the unimodal baselines for both modalities, suggesting that MASAM is effective in balanced multimodal learning.

**MASAM enables all modalities to converge jointly to flatter minima.** To better understand the advantages of MASAM in multimodal learning, we further visualize the loss landscape for the MIMIC dataset in Fig. 5 using methods presented in Li et al. (2018). Visualizations for other datasets are provided in Appendix E.2. On *both* phenotype and mortality tasks, MASAM consistently converges to flatter regions than all baseline methods for each modality encoder. While existing methods make notable progress in mitigating modality imbalance, they only rely on gradient modulation, which overlooks *the geometry of the loss landscape*. As a result, they may not ensure convergence to flat minima, leaving modality encoders prone to sharp regions and more sensitive to interference from other modalities. In contrast, MASAM overcomes these challenges by steering all modality encoders towards flatter regions of the loss landscape along their own optimization trajectories, reducing the influence of other modalities on their update directions and ensuring that each modality converges stably to a flatter solution. We provide further analysis of flat minima in Appendix F.

## 4.3 ROBUSTNESS TO LABEL NOISE AND SENSITIVITY ANALYSIS

In many real-world scenarios, label noise is often inevitable for multimodal data. We further conduct experiments by injecting different levels of label noise (20%–60%) into the CREMA-D and KS datasets. The results are summarized in Fig. 6. **MASAM consistently outperforms all baselines across varying noise levels.** This is because MASAM guides each modality to converge towards flatter minima; therefore, MASAM could achieve better generalization and robustness under noisy supervision. We conduct sensitivity analysis over key hyperparameters ($\alpha$, $\beta$, and $\rho$) on MIMIC-Phenotype and KS, with key results summarized in Appendix E.4. MASAM remains robust across a wide range of hyperparameter settings.

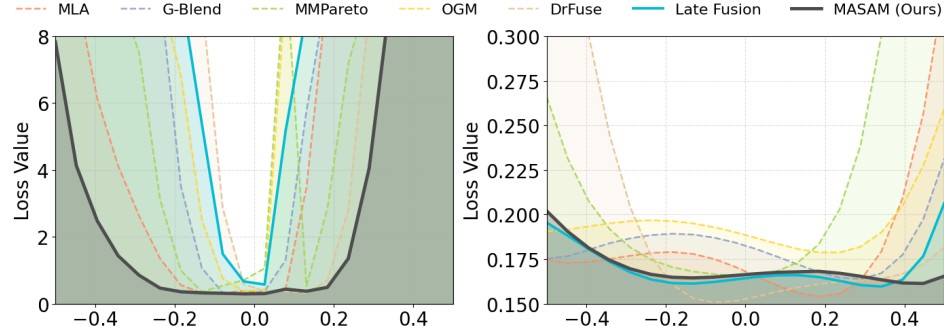

(a) The MASAM loss landscape for mortality prediction with respect to CXR (*left*) and EHR (*right*) encoders.

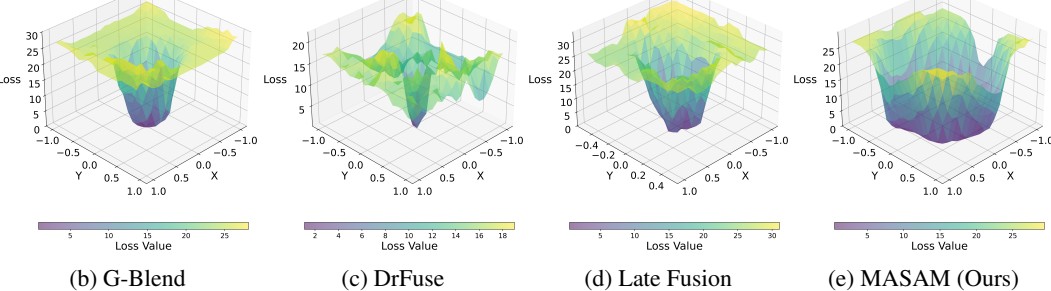

(b) G-Blend    (c) DrFuse    (d) Late Fusion    (e) MASAM (Ours)

Figure 5: Visualizations of the loss landscape. (a) 2D visualization of the MASAM loss landscape for mortality prediction. (b-e) 3D visualizations of representative methods for disease phenotype classification with respect to all model parameters. MASAM consistently achieves flat solutions jointly for both modalities.

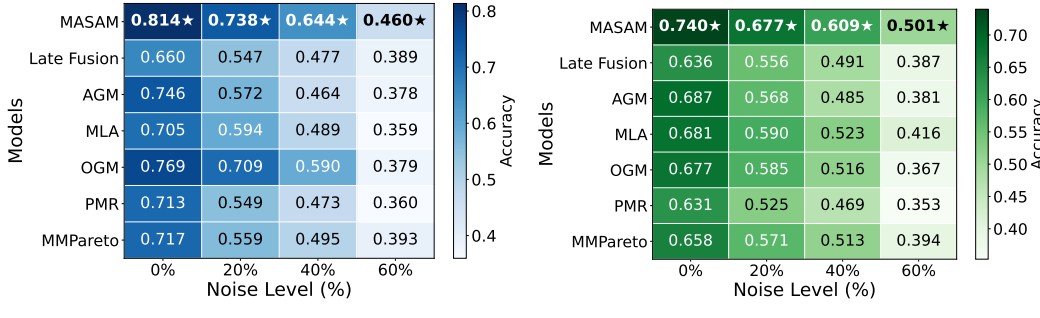

(a) CREMA-D noise robustness heatmap.    (b) KS noise robustness heatmap.

Figure 6: The performance of MASAM under varying noise levels on the CREMA-D (left) and KS (right) datasets.

## 4.4 ABLATION STUDY

To verify the contributions of each component in MASAM, we conduct an ablation study on both the MIMIC-Phenotype and KS datasets (Tables 2a and 2b) by systematically removing individual components and assessing their impact on performance. The variants include: #1 removes APS-based dynamic modality selection, #2 removes decoupled gradient perturbation, #3 employs SAM without APS or gradient decoupling, and #4 uses the standard Late Fusion strategy without sharpness-aware regularization. See Appendix E.5 for additional evaluation metrics and Appendix E.6 for comparisons of alternative APS and MDPS designs.

**(1) Adaptive perturbation score.** The APS module demonstrates a significant contribution, as shown in our ablation experiments. By comparing MASAM with #1 as well as comparing #2 with #3, we observe that APS achieves notable improvements of $2.89\%$ in MASAM and $2.72\%$ in SAM-only setting. A similar trend is observed on Kinetics-Sounds, where APS yields gains of $2.21\%$ when comparing MASAM with #1 and $4.93\%$ when comparing #2 with #3. This demonstrates that

Table 2: Ablation study of MASAM. Tables 2a and 2b report results on MIMIC-Phenotype and KS, respectively. ✓ indicates the module is enabled. Experiments are numbered for easier reference in the text. The first row shows our proposed model, while rows 1 through 4 represent different ablation experiments. All results are averaged over 4 random seeds.

(a) MIMIC-Phenotype (AUPRC)

| # | Method | APS | MDPS | SAM | AUPRC | vs. w/o APS | vs. w/o MDPS | vs. SAM Only | vs. Late Fusion |
|---|--------|-----|------|-----|-------|-------------|--------------|--------------|-----------------|
| – | MASAM | ✓ | ✓ | ✓ | **0.498** | +2.89% | +1.43% | +4.18% | +4.84% |
| 1 | w/o APS | ✗ | ✓ | ✓ | 0.484 | — | -1.43% | +1.26% | +1.89% |
| 2 | w/o MDPS | ✓ | ✗ | ✓ | 0.491 | +1.45% | — | +2.72% | +3.37% |
| 3 | SAM Only | ✗ | ✗ | ✓ | 0.478 | -1.24% | -2.65% | — | +0.63% |
| 4 | Late Fusion | ✗ | ✗ | ✗ | 0.475 | -1.86% | -3.26% | -0.63% | — |

(b) Kinetics-Sounds (Accuracy)

| # | Method | APS | MDPS | SAM | Acc. | vs. w/o APS | vs. w/o MDPS | vs. SAM Only | vs. Late Fusion |
|---|--------|-----|------|-----|------|-------------|--------------|--------------|-----------------|
| – | MASAM | ✓ | ✓ | ✓ | **0.740** | +2.21% | +2.35% | +7.40% | +16.35% |
| 1 | w/o APS | ✗ | ✓ | ✓ | 0.724 | — | +0.14% | +5.05% | +13.84% |
| 2 | w/o MDPS | ✓ | ✗ | ✓ | 0.723 | -0.14% | — | +4.93% | +13.68% |
| 3 | SAM Only | ✗ | ✗ | ✓ | 0.689 | -4.84% | -4.70% | — | +8.33% |
| 4 | Late Fusion | ✗ | ✗ | ✗ | 0.636 | -12.11% | -12.06% | -7.62% | — |

APS effectively resolves the optimization imbalance across modalities, ensuring that each modality converges to flatter regions, which leads to better overall performance.

**(2) Modality-decoupled perturbation scaling.** The decoupled gradient module also plays a critical role. Comparing MASAM with #2 and #1 with #3, we find that the decoupling mechanism provides relative improvements of $1.43\%$ in MASAM and $1.26\%$ in SAM-only setting, respectively. On Kinetics-Sounds, MDPS brings consistent benefits as well, with improvements of $2.35\%$ when comparing MASAM with #2 and $5.05\%$ when comparing #1 with #3. This affirms that our proposed MDPS plays a crucial role in effectively reducing cross-modal interference and ensuring a stable training trajectory.

**(3) Sharpness-aware minimization.** By comparing experiments 1 and 3, and 2 and 3, we observe that our APS and MDPS modules lead to performance improvements of $1.26\%$ and $2.72\%$ in the SAM-only model, respectively. On Kinetics-Sounds, combining SAM with APS and MDPS yields even larger relative gains over the SAM-only baseline (Table 2b). This demonstrates that the two proposed modules effectively address the challenges of SAM in multimodal settings: APS mitigates SAM's reliance on the dominant modality, while MDPS ensures independent optimization trajectories for each modality, avoiding the issue of forcing modalities to adapt to a shared optimization path, thus enhancing multimodal generalization.

## 5 CONCLUSION

In this paper, we propose MASAM, a multimodal adaptive sharpness-aware minimization framework to address the modality imbalance issue via attaining flat minima jointly for different modalities. At its core, MASAM dynamically identifies dominant modalities at different training stages via the adaptive perturbation score (APS), and applies modality-decoupled perturbation scaling (MDPS) to mitigate the cross-modal interference in SAM-style perturbation. This design allows each modality to preserve its own independent optimization trajectory while still benefiting from shared information. Extensive empirical evaluations across six multimodal datasets and seven downstream tasks from diverse application domains show that MASAM could attain flat minima for all modalities simultaneously, achieving balanced multimodal learning, and consequently outperforming all state-of-the-art multimodal learning methods compared. The major limitation of MASAM is that it incurs additional training overhead due to the SAM-based optimization and extra backward passes. Our future work will focus on exploring lightweight variants to improve computational efficiency while preserving performance.

## ACKNOWLEDGEMENT

This work is partially supported by the National Natural Science Foundation of China (62302413), the Health and Medical Research Fund (23220312), an Innovation and Technology Fund of Hong Kong Innovation and Technology Commission (project no ITS/202/23) and a grant under the Collaborative Research with World-leading Research Groups scheme of The Hong Kong Polytechnic University (project no G-SACF).

## ETHICS STATEMENT

All datasets used in this work are publicly available. Among them, the two datasets containing clinical data, MIMIC (Johnson et al., 2023; 2019) and ADNI (Jack Jr et al., 2008), were rigorously de-identified by the official institutions prior to release, ensuring that no personal privacy of patients is compromised. These datasets are employed strictly for research purposes. The work presented here does not pose any foreseeable risk to individuals or society.

## REPRODUCIBILITY STATEMENT

All datasets used in our experiments are publicly available. All source codes will be released, including both the model framework and the dataset processing pipeline. Additional details of the preprocessing steps are provided in Appendix A.4. For hyperparameter selection, we adopt grid search, and the best configurations for each dataset are detailed in Appendix A.3 for reproducibility.

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

# A ADDITIONAL EXPERIMENTAL DETAILS

## A.1 DATASETS AND BASELINES

This section presents the datasets used and baseline models compared in our study.

**MIMIC** (Johnson et al., 2023; 2019): MIMIC-IV contains de-identified clinical records of patients admitted to ICUs at Beth Israel Deaconess Medical Center between 2008 and 2019. Following prior works (Hayat et al., 2022; Yao et al., 2024), we extract eight important physiological variables along with the Glasgow Coma Scale (GCS) recorded within the first 48 hours of ICU stay as the EHR modality input. For the image modality, we use the most recent anterior-posterior (AP) view X-ray image acquired within the first 48 hours of ICU stays. The downstream prediction tasks include disease phenotype classification of 25 diseases and in-ICU mortality prediction. Given the inherent class imbalance in clinical data and the importance of correctly identifying positive cases, we adopt the Area Under the Precision-Recall Curve (AUPRC) as the primary evaluation metric, and report the Area Under the Receiver Operating Characteristic Curve (AUROC) as a complementary indicator.

**ADNI** (Jack Jr et al., 2008): The ADNI project has evolved through four phases, namely ADNI-1, ADNI-GO, ADNI-2, and ADNI-3, and has developed into a systematic and comprehensive multimodal research database that provides rich information for clinical prediction of Alzheimer's disease. Following the MUSE (Wu et al., 2024b) preprocessing pipeline, we selected demographic information, genetic data, and cognitive assessment scores as tabular inputs, and employed MRI scans as the visual modality. These multimodal data were utilized to perform classification across three diagnostic categories: Alzheimer's Disease (AD), Mild Cognitive Impairment (MCI), and Cognitively Normal (CN).

**CREMA-D** (Cao et al., 2014) and **Kinetics-Sounds** (Arandjelovic & Zisserman, 2017): Both datasets contain paired audio and video modalities. CREMA-D consists of audiovisual recordings of actors reading 12 English sentences under six common emotional states (angry, happy, sad, neutral, disgust, and fear), with perceptual validation provided by 2,443 crowd workers. The Kinetics dataset contains short video clips of approximately ten seconds collected from YouTube, covering a wide range of daily activities, sports, and social interactions. It is commonly used for action recognition tasks involving 31 distinct categories. Both datasets are processed in accordance with the OGM (Peng et al., 2022).

**UPMC-Food101** (Wang et al., 2015): This dataset consists of text data paired with food images. Each sample contains a food image and the corresponding web text, covering 101 distinct food categories. It is commonly used as a multimodal benchmark for food classification. We use accuracy as the primary evaluation metric.

**UR-FUNNY** (Hasan et al., 2019): This dataset contains three different modalities: text, audio, and video, and each sample is used for a binary humor classification task. We process this data following Li et al. (2023) and select accuracy as the primary evaluation metric.

## A.2 BASELINES

**Baseline methods**    We compare MASAM with the following existing methods.

- **Uni-modal**: Models trained independently on each single modality using its dedicated encoder.
- **Late Fusion**: A standard fusion strategy that concatenates modality-specific features at the decision level after separate feature extraction.
- **G-Blend** (Wang et al., 2020): It modulates the contribution of each modality during training by incorporating generation quality and overfitting sensitivity metrics.
- **AGM** (Li et al., 2023): Dynamically reweights modality gradients based on their Shapley value-estimated contributions to the overall loss.
- **MLA** (Zhang et al., 2024): It adopts a shared prediction head while alternately updating unimodal branches, enforcing directional orthogonality across modalities.
- **MMPareto** (Wei & Hu, 2024): It resolves optimization conflicts between unimodal and multimodal objectives via Pareto-based gradient projection.
- **PMR** (Fan et al., 2023): It mitigates modality imbalance by promoting weaker modalities through prototypical clustering while regularizing dominant ones with prototype-based entropy.

Table 3: Best hyperparameter configurations across datasets. $\rho$, score weight, and momentum are selected from predefined search spaces.

| Dataset | $\rho$ | Score Weight | Momentum |
|---|---|---|---|
| ADNI | 0.5 | 0.3 | 0.9 |
| CREMA-D | 0.6 | 0.3 | 0.9 |
| UPMC-Food101 | 0.3 | 0.3 | 0.9 |
| Kinetics-Sounds | 0.7 | 0.3 | 0.9 |
| MIMIC-Mortality | 0.1 | 0.3 | 0.9 |
| MIMIC-Phenotype | 0.5 | 0.3 | 0.9 |
| UR-FUNNY | 0.5 | 0.3 | 0.9 |

- **MedFuse** (Hayat et al., 2022): It sequentially aggregates multimodal representations using an LSTM-based fusion architecture, designed for partially paired clinical data.
- **DrFuse** (Yao et al., 2024): It integrates EHR and CXR modalities by disentangling modality-specific features and aligning shared representations for robust fusion.
- **AUG** (Jiang et al., 2025): It enhances modality balance from the perspective of classifier capability, using a boosting-style strategy to strengthen the weak modality's classifier.
- **InfoReg** (Huang et al., 2025): It balances the information acquisition of different modalities in the early stage by applying regularization to modalities with overly high information gain, leading to a more balanced learning process.

## A.3 SEARCH SPACE

We conducted each experiment with four random seeds ({42, 123, 1234, 2024}) and report the averaged results. Different datasets correspond to distinct search spaces, and the optimal hyperparameters obtained from grid search are reported in Table 3 to ensure the reproducibility of our results. Specifically, we search over $\rho \in \{0.1, 0.3, 0.4, 0.5, 0.6, 0.7, 0.9\}$, $\alpha \in \{0.1, 0.3, 0.5\}$, and $\beta \in \{0.5, 0.7, 0.9\}$

## A.4 CLINICAL DATA PROCESSING

### A.4.1 MIMIC DATASET

We extracted EHR and CXR data within the first 48 hours of ICU admission from MIMIC-IV (Johnson et al., 2023) and MIMIC-CXR (Johnson et al., 2019), following the preprocessing steps described in DrFuse (Yao et al., 2024) and MedFuse (Hayat et al., 2022).

For EHR data, we excluded variables with more missing ratio greater than 80%, such as those in the blood gas category. This resulted in 12 variables, including Heart Rate, Diastolic Blood Pressure, Mean Blood Pressure, Systolic Blood Pressure, Respiratory Rate, Temperature, Peripheral Capillary Oxygen Saturation, Glucose, Glasgow Coma Scale Total Score, Glasgow Coma Scale Motor Response, Glasgow Coma Scale Verbal Response, and Glasgow Coma Scale Eye Opening Response. The missing ratio information is shown in Fig. 7. We additionally added a masking column for each variable to indicate whether the value is missing at each time step, enabling the model to learn missingness-related patterns. Missing values were imputed using the median, and all variables were normalized with statistics computed from the training set. For CXR images, we used the most recent AP-view chest X-ray taken within the first 48 hours of the ICU admission.

We also analyzed the distribution of disease phenotypes in our MIMIC dataset, as shown in Table 4. The dataset contains 7,623 samples across 25 different clinical conditions, with varying positive ratios ranging from 5.26% to 49.26%. Notably, some labels exhibit substantial imbalance, which poses additional challenges for robust multimodal learning. This diverse distribution ensures comprehensive evaluation of multimodal fusion methods across different disease prevalence levels.

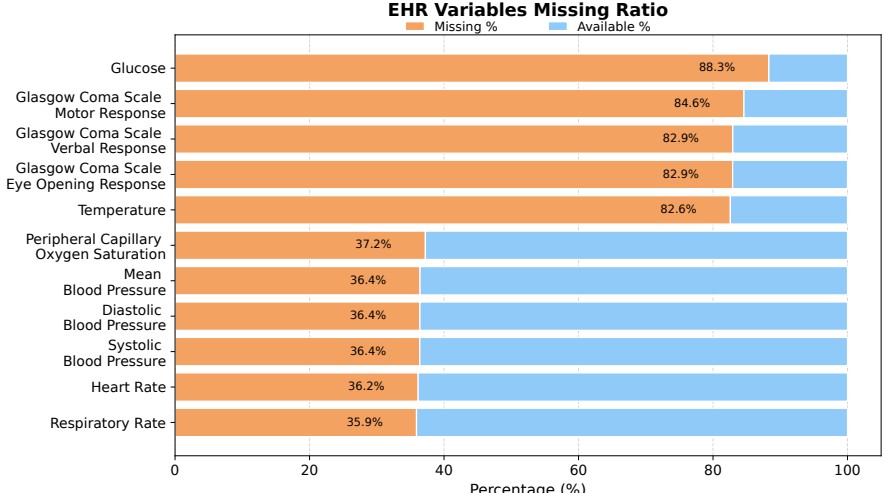

Figure 7: Missing ratios of EHR variables in our dataset. We use GCS as an abbreviation for the Glasgow Coma Scale and BP as an abbreviation for Blood Pressure. Variables with missing ratios greater than 80% were excluded from our analysis.

Table 4: Statistical overview of disease phenotypes in MIMIC dataset. The table shows the distribution of positive and negative samples across 25 clinical conditions, along with their prevalence rates.

| Disease Phenotypes | Total | Positive | Negative | Positive % |
|---|---|---|---|---|
| Fluid and electrolyte disorders | 7623 | 3755 | 3868 | 49.26 |
| Essential hypertension | 7623 | 3431 | 4192 | 45.01 |
| Disorders of lipid metabolism | 7623 | 2903 | 4720 | 38.08 |
| Cardiac dysrhythmias | 7623 | 2858 | 4765 | 37.49 |
| Acute and unspecified renal failure | 7623 | 2696 | 4927 | 35.37 |
| Respiratory failure | 7623 | 2400 | 5223 | 31.48 |
| Congestive heart failure | 7623 | 2087 | 5536 | 27.38 |
| Coronary atherosclerosis | 7623 | 1935 | 5688 | 25.38 |
| Septicemia | 7623 | 1855 | 5768 | 24.33 |
| Complications of surgical procedures | 7623 | 1668 | 5955 | 21.88 |
| Shock | 7623 | 1605 | 6018 | 21.05 |
| Pneumonia | 7623 | 1578 | 6045 | 20.70 |
| Chronic kidney disease | 7623 | 1512 | 6111 | 19.83 |
| Diabetes mellitus without complication | 7623 | 1499 | 6124 | 19.66 |
| Hypertension with complications | 7623 | 1401 | 6222 | 18.38 |
| Other liver diseases | 7623 | 1320 | 6303 | 17.32 |
| COPD and bronchiectasis | 7623 | 1159 | 6464 | 15.20 |
| Other lower respiratory disease | 7623 | 980 | 6643 | 12.86 |
| Pleurisy; pneumothorax | 7623 | 808 | 6815 | 10.60 |
| Diabetes mellitus with complications | 7623 | 794 | 6829 | 10.42 |
| Conduction disorders | 7623 | 773 | 6850 | 10.14 |
| Acute cerebrovascular disease | 7623 | 762 | 6861 | 10.00 |
| Acute myocardial infarction | 7623 | 694 | 6929 | 9.10 |
| Gastrointestinal hemorrhage | 7623 | 579 | 7044 | 7.60 |
| Other upper respiratory disease | 7623 | 401 | 7222 | 5.26 |

Table 5: Missing ratios and descriptions of variables in the ADNI dataset.

| Category | Name | Missing Ratio (%) | Description |
|---|---|---|---|
| *Demographics* | | | |
| | AGE | 0.17 | Age (years) |
| | PTGENDER | 0.00 | Gender |
| | PTEDUCAT | 0.00 | Years of education |
| | PTETHCAT | 0.00 | Ethnicity |
| | PTRACCAT | 0.00 | Race category |
| | PTMARRY | 0.04 | Marital status |
| *Genetics* | | | |
| | APOE4 | 8.64 | APOE-$\varepsilon$4 allele copies (0/1/2) |
| *CSF / PET Biomarkers* | | | |
| | ABETA | 49.77 | CSF $\beta$-amyloid protein |
| | TAU | 49.77 | CSF total Tau |
| | PTAU | 49.77 | CSF phosphorylated Tau |
| | FDG | 37.45 | FDG-PET cerebral metabolism |
| | PIB | 99.17 | PIB-PET amyloid deposition |
| | AV45 | 53.20 | AV45-PET amyloid deposition |
| | FBB | 85.78 | Florbetaben PET |
| *Cognitive Scores* | | | |
| | CDRSB | 0.00 | Clinical Dementia Rating – Sum of Boxes |
| | ADAS11 | 0.41 | Alzheimer's Disease Assessment Scale, 11 items |
| | ADAS13 | 0.91 | Alzheimer's Disease Assessment Scale, 13 items |
| | ADASQ4 | 0.17 | Alzheimer's Disease Assessment Scale, 4 items |
| | MMSE | 0.04 | Mini-Mental State Examination |
| | RAVLT_immediate | 0.33 | Rey Auditory Verbal Learning Test – Immediate recall |
| | RAVLT_learning | 0.33 | RAVLT – Learning |
| | RAVLT_forgetting | 0.37 | RAVLT – Forgetting |
| | RAVLT_perc_forget | 0.58 | RAVLT – Percent forgetting |
| | FAQ | 1.12 | Functional Activities Questionnaire |
| | MOCA | 35.26 | Montreal Cognitive Assessment |

### A.4.2 ADNI DATASET

We selected demographics, genetics, CSF/PET biomarkers, and cognitive scores as the tabular modality, while excluding variables with a high missing rate (>90%). The missing ratio of these tabular features is summarized in Table 5. For missing values within variables, we employ mean imputation and include corresponding mask indicators, while all variables are normalized using statistics computed from the training set. For prediction, we used baseline data for each subject and followed prior studies (Wu et al., 2024b; Asgharzadeh-Bonab et al., 2023; Yun et al., 2024) by merging EMCI (early mild cognitive impairment) and LMCI (late mild cognitive impairment) into a single MCI group, resulting in three classes: Alzheimer's Disease (AD), Mild Cognitive Impairment (MCI), and Cognitively Normal (CN). For the imaging modality, we adopted T1-weighted MRI scans; when a subject's baseline MRI was unavailable, we substituted scans acquired within a six-month window (Zhu et al., 2021).

## B RELATED WORK

### B.1 SHARPNESS-AWARE MINIMIZATION (SAM) AND ITS VARIANTS

Improving model generalization has long been associated with finding flatter minima in the loss landscape. SAM (Foret et al., 2021) explicitly encourages flat solutions by minimizing the worst-case loss within a neighborhood of the parameters. Building on this idea, several variants have been proposed:

- FisherSAM (Kim et al., 2022): Reformulates the parameter space as a Riemannian manifold using the Fisher information matrix. Sharpness is then measured as a distributional distance between perturbed and original parameters, offering a more principled curvature metric.
- SAGM (Wang et al., 2023): Extends SAM by simultaneously considering three objectives—empirical risk, perturbed loss, and gradient similarity—to guide optimization towards flatter and more stable regions.

- CR-SAM (Wu et al., 2024a): Observes that curvature estimation is highly sensitive to gradient scales, and introduces a scale-invariant approach to normalize curvature estimation, thus improving stability across different architectures.
- FriendlySAM (Li et al., 2024): Proposes directionally perturbed gradients, which diversify perturbation directions rather than restricting them to the normalized gradient, leading to improved generalization.

These approaches have refined the measurement of sharpness and the design of perturbations, but they are all developed in unimodal contexts. Their direct extension to multimodal learning is non-trivial because different modalities exhibit heterogeneous convergence dynamics and distinct loss landscape geometries.

## B.2 Alleviating Modality Competition in Multimodal Learning

A central challenge in multimodal learning is modality competition, where strong modalities dominate optimization, causing weaker but informative modalities to be under-optimized. This phenomenon often leads to degraded multimodal fusion and even underperformance compared to unimodal baselines (Wang et al., 2020). Several strategies have been proposed to mitigate this problem:

- Gradient modulation approaches:
  - GBlend (Wang et al., 2020): Introduces gradient blending coefficients based on the overfitting-to-generalization ratio of each modality, dynamically reweighting their contributions during training.
  - OGM (Peng et al., 2022): Promotes exploration of weaker modalities by slowing down the optimization of stronger ones.
  - AGM (Li et al., 2023): Accelerates weaker modalities by explicitly defining their relative strength and adjusting gradients accordingly.
  - PMR (Fan et al., 2023): PMR leverages prototypical representations to diagnose modality strength via prototype-based separability, regularizing strong modalities to prevent dominance and enhancing weak modalities by encouraging more discriminative prototype alignment.
- Training schedule and gradient conflict approaches:
  - MLA (Zhang et al., 2024): Proposes an iterative training strategy that alternates across modalities, combined with an orthogonality constraint on modality-specific gradients to reduce interference.
  - MMPareto (Wei & Hu, 2024): Frames multimodal optimization as a multi-objective problem and applies Pareto optimization to resolve conflicts between unimodal and multimodal gradients.
  - Diagnosing-Relearning (Wei et al., 2024): Employs a global soft reinitialization strategy, resetting modalities to varying extents during training to balance learning progress.

While these methods effectively rebalance optimization across modalities, they do so primarily by modulating gradient magnitudes or training schedules. This helps address imbalance but does not account for the loss landscape geometry of each modality. Consequently, they cannot guarantee that each modality converges to its own flat minimum, leaving the generalization benefits of flatness-aware optimization unexplored.

In summary, prior work on SAM and its variants focuses on **flatness for unimodal tasks**, while multimodal learning methods emphasize **gradient-based balancing** across modalities. However, no existing method explicitly addresses how **sharpness-aware optimization interacts with multimodal imbalance**. This motivates our proposed MASAM framework, which integrates flatness-aware perturbation with modality-specific adaptation to overcome the limitations of both lines of research.

## C Derivation of Observation 1

Without loss of generality, here we assume late fusion of two modalities. In the following content we continue to use notations in Section 3.1. In late fusion methods, although each modality appears to be optimized independently, their training remains inherently coupled through the shared fused prediction. Specifically, the gradient of the fused loss with respect to the parameters of modality $m$

can be expressed as:

$$\nabla_{\boldsymbol{\theta}_m} \mathcal{L}_{\text{fuse}} = \sum_{i=1}^{N} \sum_{c=1}^{C} \frac{\partial \mathcal{L}_{\text{fuse}}}{\partial \hat{y}_{i,c}^{\text{fuse}}} \cdot \frac{\partial \hat{y}_{i,c}^{\text{fuse}}}{\partial \mathbf{z}_i^{\text{fuse}}} \cdot \frac{\partial \mathbf{z}_i^{\text{fuse}}}{\partial \mathbf{z}_i^m} \cdot \frac{\partial \mathbf{z}_i^m}{\partial \boldsymbol{\theta}_m} = \sum_{i=1}^{N} \sum_{c=1}^{C} (\hat{y}_{i,c}^{\text{fuse}} - y_{i,c}) \cdot \mathbf{W}_{c,:}^m \cdot \frac{\partial \phi_m(x_i^m)}{\partial \boldsymbol{\theta}_m}. \quad (15)$$

Here, the fused prediction is defined as $\hat{y}_{i,c}^{\text{fuse}} = g\left(\mathbf{W}_{c,:}^{m_1} \cdot \mathbf{z}_i^{m_1} + \mathbf{W}_{c,:}^{m_2} \cdot \mathbf{z}_i^{m_2} + b_c\right)$, where $g(\cdot)$ denotes the task-specific activation function: a softmax function in multiclass classification tasks with categorical cross-entropy loss, or a sigmoid function in multilabel tasks with binary cross-entropy loss. This formulation follows the standard late fusion paradigm, where the fused weight vector $\mathbf{W}_{c,:}$ in Eq. (4) can be decomposed into modality-specific parts $\mathbf{W}_{c,:}^{m_1}$ and $\mathbf{W}_{c,:}^{m_2}$. From this formulation, it is evident that a dominant modality (e.g., $m_2$) with stronger discriminative signals will exert more influence on the final prediction through a larger weighted contribution ($\mathbf{W}_{c,:}^{m_2} \cdot \mathbf{z}_i^{m_2}$). This dominance causes the computed loss gradient to be skewed in favor of the stronger modality, while the weaker modality (e.g., $m_1$) receives gradients that are misaligned with its own representation, thereby limiting its learning capacity and reducing effective exploration.

This issue also extends to the computation of the SAM perturbation Eq. (3). Since the gradient itself is already biased by the dominant modality, the resulting perturbation vector for the weaker modality becomes largely determined by the optimization direction favored by the strong modality. As a result, this perturbation may not only fail to benefit the weaker modality but may even steer it in a suboptimal or harmful direction, thereby undermining the effectiveness of SAM and exacerbating the imbalance across modalities.

## D    PROOF OF THEOREM 1

We now provide a full proof of Theorem 1 using the framework of inexact gradient descent (IGD). Let $\widetilde{\mathbf{g}}_m^t = \frac{\nabla_{\boldsymbol{\theta}_m} \mathcal{L}_{\text{fuse}}^t}{\left\| \nabla_{\boldsymbol{\theta}_m} \mathcal{L}_{\text{fuse}}^t \right\|_2}$ denote the normalized gradient of the fusion objective with respect to the encoder parameters of modality $m$ at step $t$.

**Lemma 1** (Perturbation Error Bound for Fusion Gradient). *For each modality $m$, given the $L_{\text{fuse}}$-Lipschitz continuity of $\nabla \mathcal{L}_{\text{fuse}}$, the perturbed gradient of $\mathcal{L}_{\text{fuse}}$ satisfies*

$$\left\| \nabla_{\theta_m} \mathcal{L}_{\text{fuse}}(\theta_m^t + \rho_t \, \gamma_m^t \, \widetilde{\mathbf{g}}_m^t) - \nabla_{\theta_m} \mathcal{L}_{\text{fuse}}(\theta_m^t) \right\| \le L_{\text{fuse}} \rho_t. \quad (16)$$

*Proof.* By Lipschitz continuity of $\nabla \mathcal{L}_{\text{fuse}}$, i.e.,

$$\left\| \nabla_{\theta_m} \mathcal{L}_{\text{fuse}}(\theta_m^t + \rho_t \, \gamma_m^t \, \widetilde{\mathbf{g}}_m^t) - \nabla_{\theta_m} \mathcal{L}_{\text{fuse}}(\theta_m^t) \right\| \le L_{\text{fuse}} \| \rho_t \, \gamma_m^t \, \widetilde{\mathbf{g}}^t \|.$$

Eq. (16) is obtained as $\|\gamma_m^t\| \le 1$ and $\|\widetilde{\mathbf{g}}_m^t\| = 1$.  $\square$

**Lemma 2** (MASAM as IGD instance). *The MASAM update (Eq. (12)) for modality $m$ is an instance of the inexact gradient descent method as:*

$$\theta_m^{t+1} = \theta_m^t - \eta_t g_m^t \text{ with inexact condition } \|g_m^t - \nabla_{\theta_m} \mathcal{L}_{total}(\theta_m^t)\| \le L_{\text{fuse}} \rho_t. \quad (17)$$

*Proof.* Let $g_m^t := \nabla_{\theta_m} \mathcal{L}_{\text{fuse}}(\theta_m^t + \rho_t \, \gamma_m^t \, \widetilde{\mathbf{g}}_m^t) + \nabla_{\theta_m} \mathcal{L}_m(\theta_m^t)$. Then the MASAM update gives $\theta_m^{t+1} = \theta_m^t - \eta_t g_m^t$. Together with Eq. (7) and Eq. (16), the inexact condition can be obtained as:

$$\|g_m^t - \nabla_{\theta_m} \mathcal{L}_{\text{total}}(\theta_m^t)\| = \left\| \nabla_{\theta_m} \mathcal{L}_{\text{fuse}}(\theta_m^t + \rho_t \, \gamma_m^t \, \widetilde{\mathbf{g}}_m^t) + \nabla_{\theta_m} \mathcal{L}_m(\theta_m^t) - \nabla_{\theta_m} \mathcal{L}_{\text{total}}(\theta_m^t) \right\|$$

$$= \left\| \nabla_{\theta_m} \mathcal{L}_{\text{fuse}}(\theta_m^t + \rho_t \, \gamma_m^t \, \widetilde{\mathbf{g}}_m^t) + \nabla_{\theta_m} \mathcal{L}_m(\theta_m^t) - \nabla_{\theta_m} \mathcal{L}_{\text{fuse}}(\theta_m^t) - \nabla_{\theta_m} \mathcal{L}_m(\theta_m^t) \right\|$$

$$= \left\| \nabla_{\theta_m} \mathcal{L}_{\text{fuse}}(\theta_m^t + \rho_t \, \gamma_m^t \, \widetilde{\mathbf{g}}_m^t) - \nabla_{\theta_m} \mathcal{L}_{\text{fuse}}(\theta_m^t) \right\| \le L_{\text{fuse}} \rho_t.$$

$\square$

We next provide the proof of Theorem 1 based on Lemma 1, Lemma 2, and Khanh et al. (2024, Theorem 3.3).

Table 6: This table compares the AUPRC of different models across 25 disease phenotypes. Bold values indicate the best performance, and underlined values indicate second-best performance. The Average Rank metric ranks all models for each phenotype individually and calculates their mean ranking across all 25 disease labels. MASAM consistently ranked first or second across nearly all phenotypes and achieved the highest average rank overall, validating the effectiveness of our approach.

| Clinical Condition | Uni-EHR | Uni-CXR | G-Blend | OGM | AGM | MLA | MMPareto | DrFuse | MASAM |
|---|---|---|---|---|---|---|---|---|---|
| Acute renal failure | 0.593 | 0.519 | 0.590 | 0.577 | 0.578 | 0.587 | 0.590 | 0.594 | **0.599** |
| Acute cerebrovascular disease | 0.465 | 0.193 | 0.491 | 0.484 | **0.500** | 0.471 | 0.478 | 0.496 | 0.487 |
| Acute myocardial infarction | 0.205 | 0.188 | 0.243 | 0.245 | 0.236 | 0.230 | 0.248 | **0.577** | 0.253 |
| Cardiac dysrhythmias | 0.576 | 0.583 | 0.638 | 0.633 | 0.627 | 0.625 | 0.634 | 0.412 | **0.653** |
| Chronic kidney disease | 0.472 | 0.461 | 0.550 | 0.535 | 0.528 | 0.542 | 0.539 | **0.581** | 0.556 |
| COPD and bronchiectasis | 0.328 | 0.361 | 0.424 | 0.413 | 0.407 | 0.403 | 0.420 | **0.572** | 0.435 |
| Surgical complications | 0.331 | 0.286 | 0.344 | 0.338 | 0.335 | 0.335 | 0.333 | **0.690** | 0.335 |
| Conduction disorders | 0.214 | 0.597 | 0.615 | 0.626 | 0.618 | 0.616 | 0.621 | 0.199 | **0.634** |
| CHF; nonhypertensive | 0.548 | 0.671 | 0.703 | 0.691 | 0.682 | 0.689 | 0.699 | 0.506 | **0.719** |
| CAD | 0.472 | 0.579 | 0.610 | 0.598 | 0.605 | 0.588 | 0.599 | 0.417 | **0.621** |
| DM with complications | 0.591 | 0.238 | 0.571 | 0.566 | 0.556 | 0.559 | 0.577 | 0.177 | **0.587** |
| DM without complication | 0.380 | 0.295 | 0.400 | 0.395 | 0.395 | 0.393 | **0.404** | 0.207 | 0.407 |
| Disorders of lipid metabolism | 0.563 | 0.551 | 0.585 | 0.575 | 0.573 | 0.584 | 0.578 | 0.241 | **0.586** |
| Essential hypertension | 0.542 | 0.524 | 0.566 | 0.565 | 0.565 | 0.568 | 0.562 | 0.217 | **0.576** |
| Fluid and electrolyte disorders | 0.678 | 0.625 | **0.689** | 0.677 | 0.676 | 0.681 | 0.687 | 0.436 | 0.686 |
| Gastrointestinal hemorrhage | 0.171 | 0.168 | 0.199 | 0.200 | 0.188 | 0.203 | 0.204 | **0.638** | 0.225 |
| Secondary hypertension | 0.446 | 0.424 | 0.506 | 0.494 | 0.483 | 0.492 | 0.503 | **0.572** | 0.511 |
| Other liver diseases | 0.309 | 0.399 | 0.419 | 0.415 | 0.395 | 0.406 | 0.418 | **0.591** | 0.445 |
| Other lower respiratory disease | 0.160 | 0.191 | 0.178 | 0.178 | 0.181 | 0.173 | 0.183 | **0.638** | 0.185 |
| Other upper respiratory disease | 0.206 | 0.161 | 0.223 | 0.247 | 0.219 | 0.265 | 0.238 | **0.544** | 0.309 |
| Pleurisy; pneumothorax | 0.134 | 0.238 | 0.232 | 0.230 | 0.215 | 0.210 | 0.229 | **0.415** | 0.235 |
| Pneumonia | 0.392 | 0.392 | 0.438 | 0.431 | 0.428 | 0.433 | 0.434 | 0.335 | **0.441** |
| Respiratory failure | 0.612 | 0.564 | 0.644 | 0.637 | 0.642 | 0.643 | 0.643 | 0.617 | **0.650** |
| Septicemia | 0.569 | 0.431 | 0.566 | 0.565 | 0.567 | 0.565 | 0.565 | **0.699** | 0.576 |
| Shock | 0.571 | 0.440 | 0.581 | 0.572 | 0.579 | 0.589 | 0.590 | **0.608** | 0.594 |
| **Average Rank** | 6.38 | 6.72 | 4.36 | 4.98 | 5.16 | 5.16 | 4.72 | 4.16 | **2.36** |

*Proof of Theorem 1.* Given that Khanh et al. (2024, Theorem 3.3) already established the conditions for IGD method and that the IGD format of MASAM is given in Eq. (17), the convergence of MASAM requires that

$$\text{(i) } \sum_{t=1}^{\infty} \eta_t = \infty, \quad \text{(ii) } \eta_t \downarrow 0, \quad \text{(iii) } \sum_{t=1}^{\infty} \eta_t \, L_{\text{fuse}} \, \rho_t < \infty, \quad \text{(iv) } \limsup_{t \to \infty} L_{\text{fuse}} \rho_t < 2, \quad (18)$$

which is equivalent to the conditions in Eq. (14). Then the convergence of MASAM can be obtained. □

# E  ADDITIONAL EXPERIMENT RESULTS

## E.1  PREDICTION PERFORMANCE ACROSS DISEASE PHENOTYPES

We report the detailed breakdown of prediction performance across different disease phenotypes in Table Table 6.

## E.2  LOSS LANDSCAPE VISUALIZATION

We visualize the loss landscapes across different baseline models for all datasets: MIMIC-Phenotype (Fig. 8), MIMIC-Mortality (Fig. 9), CREMA-D (Fig. 10), Kinetics-Sounds (Fig. 11), UPMC-Food101 (Fig. 12), and ADNI (Fig. 13). Across all these figures, we clearly observe that our proposed MASAM method converges to a noticeably flatter region than all other models, demonstrating stronger generalization and robustness.

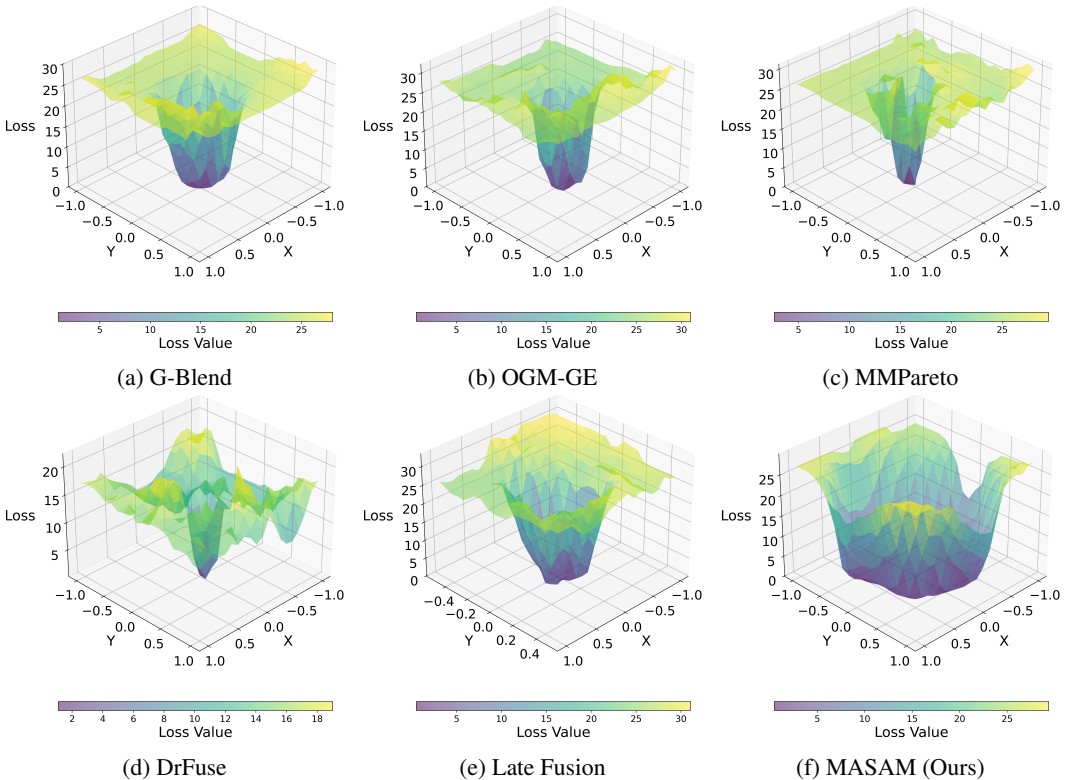

Figure 8: 3D loss landscape visualization of MASAM and baseline models in disease phenotype classification. MASAM consistently converges to a wider and flatter region, indicating better generalization and robustness.

### E.3 ADDITIONAL METIRCS

In this section, we provide additional experimental results to complement the main findings. Specifically, we report AUROC for MIMIC-Phenotype and MIMIC-Mortality, mAP for CREMA-D, Food101, and KS, and Accuracy for ADNI. These results offer a more comprehensive evaluation of the effectiveness of our proposed method.

As shown in Table 7, MASAM consistently achieves notable improvements across diverse datasets and tasks on these supplementary metrics, providing strong evidence of the effectiveness of our approach.

### E.4 SENSITIVE ANALYSIS

To assess the robustness of our proposed model with respect to hyperparameter choices and to gain insights into the functional roles of key components in our design, we perform a sensitivity analysis on three key parameters: $\alpha, \beta, \rho$. The parameter $\alpha$ controls the weighting between the loss descent rate and gradient alignment in APS computation. A higher $\alpha$ emphasizes the contribution of the loss descent rate when estimating modality dominance. The parameter $\beta$ determines the momentum accumulation rate for loss smoothing; a larger $\beta$ places more emphasis on the historical trend of loss values, making the model's judgment of descent more reliant on long-term behavior, whereas a smaller $\beta$ increases sensitivity to short-term fluctuations. The parameter $\rho$ specifies the magnitude of the perturbation step in sharpness-aware optimization. A larger $\rho$ expands the neighborhood over which the loss surface is explored, effectively enlarging the search space.

MASAM demonstrates strong robustness to the hyperparameter $\alpha$, while also highlighting the importance of gradient alignment in quantifying dominant modalities. As shown in Fig. 14a, MASAM consistently outperforms Late Fusion across a wide range of $\alpha$ values (0.1 to 0.8). Notably, the

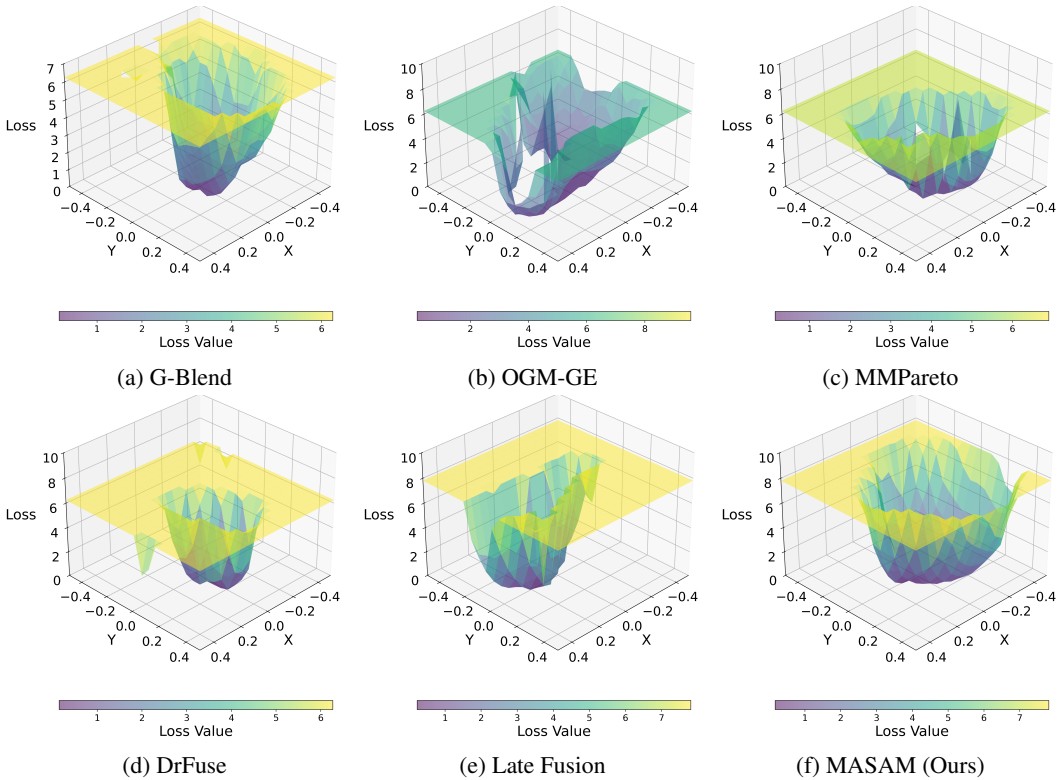

Figure 9: 3D loss landscape visualization of MASAM and baseline models in mortality prediction task. MASAM converges to a wider and flatter region, indicating improved robustness and generalization compared to all baselines.

model achieves the best performance when $\alpha$ is set between 0.2 and 0.3, suggesting that the degree of gradient alignment plays a more critical role than loss descent rate in estimating modality dominance in our settings. A similar trend is observed on the KSdataset (see Fig. 14d), where MASAM again reaches its peak performance around $\alpha = 0.2$–$0.3$, further confirming the central role of gradient alignment in estimating modality dominance across diverse multimodal settings.

Similarly, MASAM exhibits strong robustness with respect to the momentum parameter $\beta$, further confirming that incorporating global loss trends is beneficial for estimating the learning speed of each modality. As shown in Fig. 14b, MASAM maintains consistently high AUPRC scores across a broad range of $\beta$ values (0.1 to 0.7), with a slight increase observed at $\beta = 0.9$, where the model achieves its best performance. A similar pattern is observed on the KSdataset (see Fig. 14e), where the model likewise attains its highest accuracy at $\beta = 0.9$, further reinforcing that emphasizing long-term loss trends enhances the reliability of modality-wise learning estimation across datasets. Overall, the performance remains stable and significantly superior to that of Late Fusion, demonstrating that leveraging global loss dynamics helps stabilize modality-wise learning assessments.

MASAM continues to demonstrate strong robustness with respect to the perturbation magnitude $\rho$. As shown in Fig. 14c, the MIMIC-Phenotype dataset exhibits a mild dip at $\rho = 0.6$, followed by a recovery as $\rho$ increases. In contrast, the KS dataset (Fig. 14f) displays a steadily increasing trend throughout the entire range, with the best performance achieved at larger perturbation radii ($\rho = 0.7$–$0.9$). These observations suggest that the optimal perturbation magnitude can vary across datasets, reflecting differences in loss landscape geometry. Although the KS curve shows more noticeable variation in the small-$\rho$ region (0.1–0.3), the performance becomes stable and consistently strong for $\rho \geq 0.3$, supporting the robustness of MASAM under a wide range of perturbation scales.

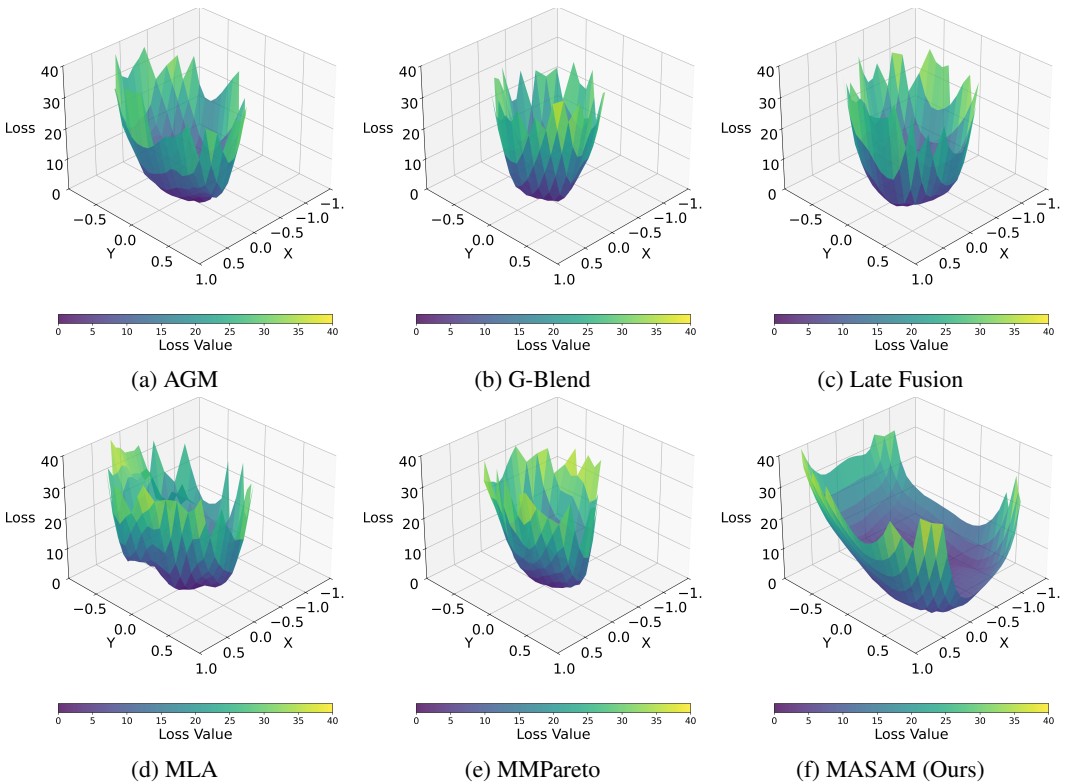

Figure 10: 3D loss landscape visualization of MASAM and baseline models in CREMA-D. MASAM consistently converges to a wider and flatter region, indicating better generalization and robustness.

### E.5 ABLATION STUDY RESULTS MEASURED BY AUROC

We provide the results of ablation studies in AUROC in Table 9.

### E.6 APS AND MDPS ALTERNATIVES

To assess whether our design choices are necessary, we evaluate several alternative formulations for APS and MDPS. For APS, we test: (1) replacing our score with the modality–scoring mechanism used in OGM (Peng et al., 2022); and (2) setting $\alpha = 0$ in APS, which removes the loss–decay term and keeps only the gradient-alignment component $\gamma_m^{(t)}$. Table 8 summarizes the comparison between MASAM and the alternative designs. For MDPS, we explore two alternative perturbation directions: (1) using the gradient difference between fusion and unimodal objectives, $\mathbf{g}'_1 = \nabla_{\boldsymbol{\theta}_{\mathrm{m}}}\mathcal{L}_{\mathrm{fuse}} - \nabla_{\boldsymbol{\theta}_{\mathrm{m}}}\mathcal{L}_{\mathrm{m}}$ (2) using the orthogonal component of the unimodal gradient, $\mathbf{g}'_2 = \nabla_{\boldsymbol{\theta}_{\mathrm{m}}}\mathcal{L}_{\mathrm{fuse}}$. Table 8 summarizes the comparison between MASAM and the alternative designs. We first examine the APS variants (#1 and #2). Using the OGM scoring rule reduces performance on KS by $1.22\%$ and on MIMIC-Phenotype by $1.01\%$. Removing the loss-decay term by setting $\alpha = 0$ also causes drops of $0.95\%$ on KS and $1.81\%$ on MIMIC-Phenotype. These results indicate that the APS alternatives do not reliably identify the dominant modality for SAM optimization. Next, we evaluate the MDPS variants (#3 and #4). The gradient-difference variant $\mathbf{g}'_1$ leads to a much larger degradation, with drops of $5.41\%$ on KS and $7.23\%$ on MIMIC-Phenotype. The orthogonal-gradient variant $\mathbf{g}'_2$ also performs worse than MASAM on both datasets. This shows that these alternatives cannot effectively decouple cross-modal interference during SAM perturbation.

### E.7 GRADIENT ALIGNMENT ANALYSIS

Fig. 15 reports the cosine similarity between the unimodal and fusion gradients for the audio-video encoders in KS and the EHR-CXR in MIMIC-Phenotype over training epochs. Across all modalities,

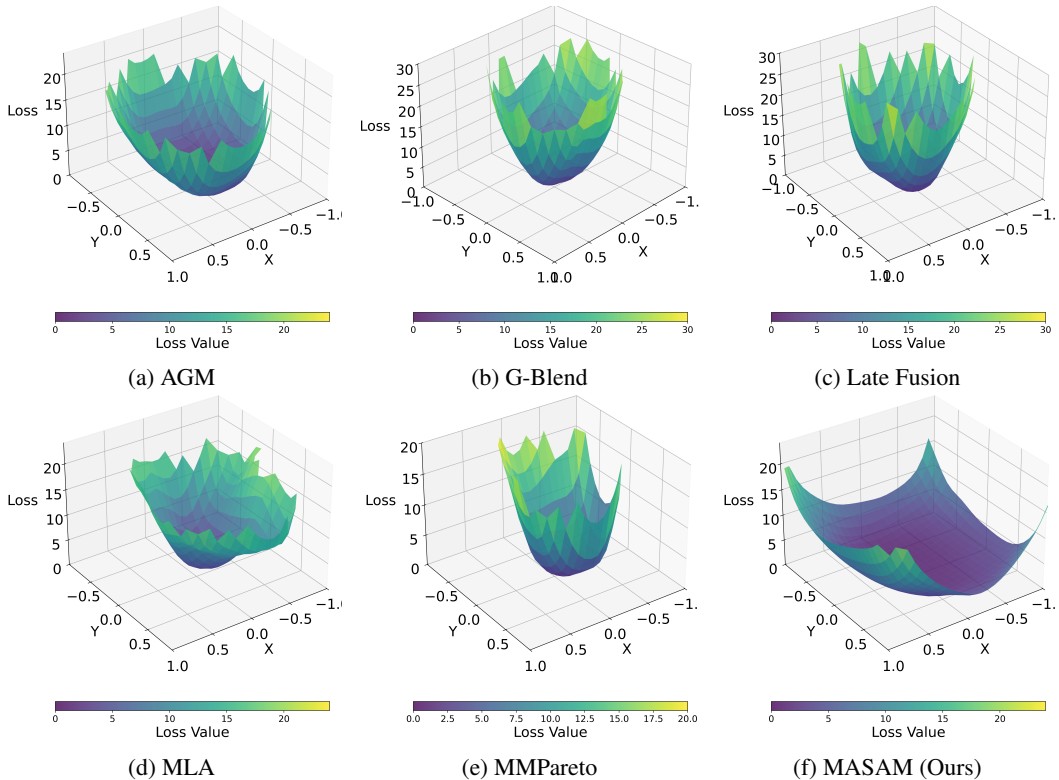

Figure 11: 3D loss landscape visualization of MASAM and baseline models on the Kinetics-Sounds dataset. MASAM converges to a wider and flatter region, demonstrating improved generalization and robustness.

MASAM consistently maintains higher and more stable alignment than all baselines, and the gap widens as training progresses. This empirically supports that our MDPS design effectively aligns unimodal and fusion objectives, leading to more coherent multimodal optimization.

### E.8 EXTENSION TO TRI-MODALITY DATASET

We further demonstrate the scalability of MASAM by extending it to a three-modality setting on the UR-FUNNY (Hasan et al., 2019) and comparing it with baselines that also support this setting. To extend MASAM to the three-modality setting, we compute the modality-specific $\text{APS}_m^{(t)}$ for all modalities at each training step and identify the dominant modality $m^* = \arg\max_m \text{APS}_m^{(t)}$. MASAM then applies the MDPS-guided SAM perturbation *only* to the dominant modality $m^*$, while the remaining modalities proceed with their standard gradient updates.

## F FLAT MINIMA ANALYSIS

Gradient scaling methods are effective in solving modality competition, but they cannot ensure that each modality converges to a flat minima. In this part, we provide an in-depth analysis.

These works (Li et al., 2023; Peng et al., 2022; Wang et al., 2020) adopt gradient-scaling strategies, where each modality's gradient is scaled by a static coefficient $c$ based on a modality importance score. This leads to an update rule of the form:

$$\theta^{t+1} = \theta^t - \eta_t \, c \, \nabla \mathcal{L}(\theta^t).$$ (19)

The flatness of the loss landscape is often correlated with the largest eigenvalue of the Hessian matrix, $\lambda_{\max}(H(w))$ (Yang et al., 2021; Sankar et al., 2021; Wu et al., 2024a). To analyze how the update

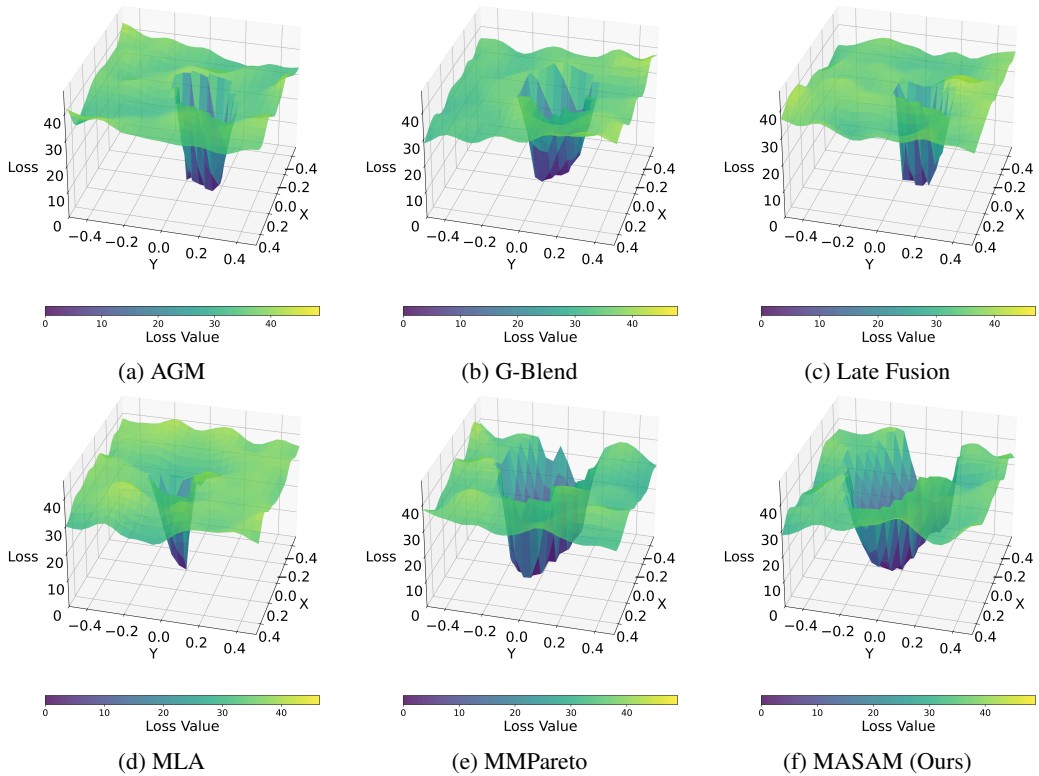

Figure 12: 3D loss landscape visualization of MASAM and baseline models on the UPMC-Food101 dataset. MASAM converges to a wider and flatter region, demonstrating improved generalization and robustness.

affects this flatness, we apply a first-order Taylor expansion of the Hessian around $w$:

$$H(\theta^{t+1}) \;\approx\; H(\theta^t) - \eta_t c \, \nabla H(\theta^t)[\nabla \mathcal{L}(\theta^t)]. \tag{20}$$

Since the Hessian is symmetric, we approximate the change in its largest eigenvalue using the Rayleigh quotient:

$$\lambda_{\max}(H(\theta^{t+1})) \;\approx\; \lambda_{\max}(H(\theta^t)) - \eta_t c \cdot \left[v^\top \nabla H(\theta^t)[\nabla \mathcal{L}(\theta^t)]v\right], \tag{21}$$

where $v$ is the eigenvector corresponding to $\lambda_{\max}(H(\theta^t))$.

This analysis shows that simply scaling the gradient by a positive coefficient $c$ does not alter the direction of the term $\nabla H(w)[\nabla L(w)]$, which governs whether the model moves towards sharper or flatter regions. The effectiveness of such an update in reducing sharpness depends on the natural geometry of the optimization landscape, not on the algorithm itself.

In multimodal settings, this limitation is especially critical: a modality that initially converges towards a flatter region may be pulled into sharper areas due to the continued influence of other modalities. Gradient-scaling methods lack explicit mechanisms to prevent this, whereas MASAM explicitly regularizes each modality towards flatter minima, thereby mitigating cross-modality interference and ensuring more stable optimization.

## G  COMPUTATIONAL EFFICIENCY ANALYSIS

We evaluate whether MASAM incurs additional computational overhead by measuring the average per-step inference latency training latency and the peak GPU memory usage under the best-performing configuration on the MIMIC-Phenotype prediction task, the results are reported in Table 10. MASAM achieves an inference per-step latency (8.875 ms) comparable to Late Fusion (8.637 ms), indicating

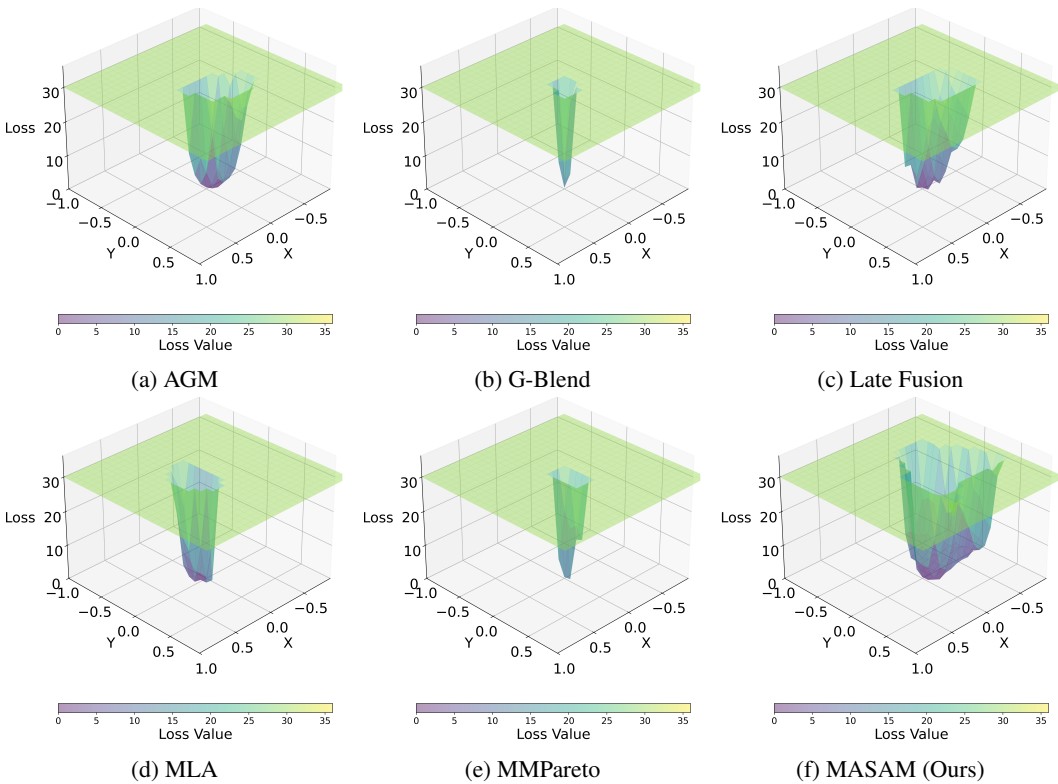

Figure 13: 3D loss landscape visualization of MASAM and baseline models on the ADNI dataset. MASAM converges to a substantially flatter and broader region, illustrating enhanced robustness and generalization.

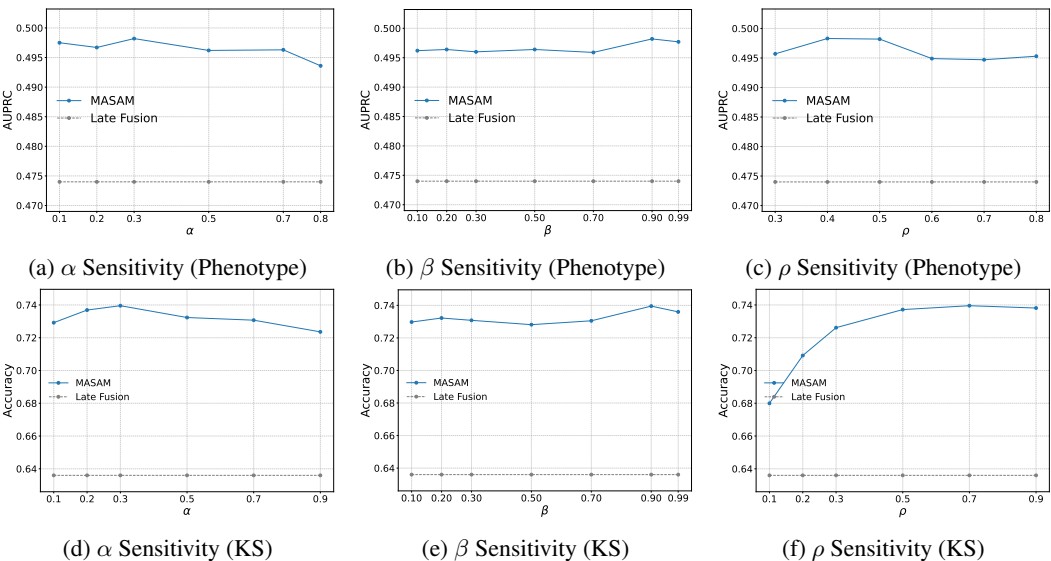

Figure 14: Sensitivity analysis of MASAM on three key hyperparameters: $\alpha$, $\beta$, and $\rho$ with accuracy as the metric. The first row reports results on the MIMIC-Phenotype dataset, while the second row presents the corresponding results on Kinetics-Sounds (KS). All numbers represent averages across 4 random seeds.

that our method does not introduce noticeable overhead at inference time. While MASAM requires a moderate amount of additional memory due to SAM-based optimization, its peak usage remains significantly lower than computationally intensive baseline DrFuse. However, our method exhibits a

Table 7: Unified comparison on additional metrics. Columns report AUROC for MIMIC-Phenotype and MIMIC-Mortality, mAP f r CREMA-D and KS, and Accuracy for ADNI. Underlined values indicate the strongest baseline, while bold highlights our method. Relative Gain is computed against the strongest baseline in each column. All results are averaged over four runs with different random seeds.

| | Phenotype | Mortality | CREMA-D | KS | ADNI |
|---|---|---|---|---|---|
| **Method** | AUROC | AUROC | mAP | mAP | Accuracy |
| Late Fusion | $0.748_{\pm 0.0023}$ | $0.903_{\pm 0.0062}$ | $0.660_{\pm 0.0209}$ | $0.678_{\pm 0.0120}$ | $0.749_{\pm 0.0047}$ |
| G-Blend (Wang et al., 2020) | $0.752_{\pm 0.0010}$ | $0.902_{\pm 0.0008}$ | $0.717_{\pm 0.0134}$ | $0.710_{\pm 0.0109}$ | $0.751_{\pm 0.0064}$ |
| OGM (Peng et al., 2022) | $0.748_{\pm 0.0021}$ | $0.906_{\pm 0.0015}$ | $0.769_{\pm 0.0040}$ | $0.729_{\pm 0.0049}$ | $0.756_{\pm 0.0133}$ |
| AGM (Li et al., 2023) | $0.746_{\pm 0.0021}$ | $0.906_{\pm 0.0034}$ | $0.746_{\pm 0.0106}$ | $0.747_{\pm 0.0052}$ | $0.757_{\pm 0.0111}$ |
| MLA (Zhang et al., 2024) | $0.748_{\pm 0.0019}$ | $\underline{0.907}_{\pm 0.0005}$ | $0.705_{\pm 0.0033}$ | $0.723_{\pm 0.0121}$ | $0.755_{\pm 0.0158}$ |
| MMPareto (Wei & Hu, 2024) | $0.751_{\pm 0.0006}$ | $0.904_{\pm 0.0052}$ | $0.717_{\pm 0.0106}$ | $0.712_{\pm 0.0076}$ | $0.751_{\pm 0.0098}$ |
| PMR (Fan et al., 2023) | N.A.[†] | $0.906_{\pm 0.0018}$ | $0.782_{\pm 0.0370}$ | $0.665_{\pm 0.0120}$ | $0.755_{\pm 0.0110}$ |
| InfoReg | $\underline{0.753}_{\pm 0.0008}$ | $0.904_{\pm 0.0035}$ | $0.792_{\pm 0.0190}$ | $0.734_{\pm 0.0065}$ | $0.753_{\pm 0.0120}$ |
| AUG | $0.747_{\pm 0.0018}$ | $0.901_{\pm 0.0079}$ | $\underline{0.838}_{\pm 0.0167}$ | $\underline{0.752}_{\pm 0.0106}$ | $0.755_{\pm 0.0142}$ |
| DrFuse (Yao et al., 2024) | $0.749_{\pm 0.0003}$ | $0.905_{\pm 0.0011}$ | N.A.[‡] | N.A.[‡] | $\underline{0.768}_{\pm 0.0058}$ |
| MedFuse (Hayat et al., 2022) | $0.747_{\pm 0.0021}$ | $0.895_{\pm 0.0010}$ | N.A.[‡] | N.A.[‡] | $0.760_{\pm 0.0156}$ |
| **MASAM (ours)** | $\mathbf{0.760}_{\pm 0.0004}$ | $\mathbf{0.915}_{\pm 0.0010}$ | $\mathbf{0.797}_{\pm 0.0091}$ | $\mathbf{0.797}_{\pm 0.0037}$ | $\mathbf{0.778}_{\pm 0.0076}$ |
| *Relative Gain (%)* | *+0.93* | *+0.88* | *-4.89* | *+5.98* | *+1.30* |

[†] PMR relies on a contrastive module that is incompatible with multi-label phenotype prediction.

[‡] DrFuse and MedFuse are tailored for clinical data; thus, we exclude them on CREMA-D and KS.

Table 8: Comparison of alternative APS and MDPS designs on both KS and MIMIC-Phenotype datasets. ✓ indicates the default design in MASAM. All results are averaged over four random seeds.

| # | Method | KS Accuracy | $\Delta$ Acc (%) | MIMIC-Phenotype AUPRC | $\Delta$ AUPRC (%) |
|---|---|---|---|---|---|
| – | MASAM (Default) | $\mathbf{0.740} \pm 0.0084$ | 0.00% | $\mathbf{0.498} \pm 0.0010$ | 0.00% |
| 1 | MASAM w/ OGM Score | $0.731 \pm 0.0012$ | -1.22% | $0.493 \pm 0.0013$ | -1.00% |
| 2 | MASAM w/ APS ($\alpha = 0$) | $0.733 \pm 0.0043$ | -0.95% | $0.489 \pm 0.0033$ | -1.81% |
| 3 | MASAM w/ $\mathbf{g}_1'$ | $0.700 \pm 0.0064$ | -5.41% | $0.462 \pm 0.0026$ | -7.23% |
| 4 | MASAM w/ $\mathbf{g}_2'$ | $0.721 \pm 0.0074$ | -2.57% | $0.489 \pm 0.0010$ | -1.81% |

Table 9: Ablation study of MASAM in disease phenotypes classification task (AUROC). ✓ indicates the module is enabled. Experiments are numbered for easier reference in the text. The first row shows our proposed model, while rows 1 through 4 represent different ablation experiments. All results are averaged over 4 random seeds. This table demonstrates the ablation experiment results using the AUROC metric, revealing that both our proposed APS and MDPS modules make significant contributions. Comparing MASAM with #1, #2, and #3 shows that the APS module substantially improves model performance. Similarly, comparisons between MASAM and #2, as well as between #1 and #3, demonstrate that the MDPS module enhances the model's overall effectiveness.

| # | Method | APS | MDPS | SAM | AUROC | *vs. w/o APS* | *vs. w/o MDPS* | *vs. SAM Only* | *vs. Late Fusion* |
|---|---|---|---|---|---|---|---|---|---|
| – | MASAM | ✓ | ✓ | ✓ | **0.760** | +0.93% | +0.26% | +1.06% | +1.60% |
| 1 | w/o APS | ✗ | ✓ | ✓ | 0.753 | — | -0.66% | +0.13% | +0.67% |
| 2 | w/o MDPS | ✓ | ✗ | ✓ | 0.758 | +0.27% | — | +0.80% | +1.34% |
| 3 | SAM Only | ✗ | ✗ | ✓ | 0.752 | -0.13% | -0.79% | — | +0.53% |
| 4 | Late Fusion | ✗ | ✗ | ✗ | 0.748 | -0.66% | -1.32% | -0.53% | — |

longer average training time due to the additional backpropagation required by SAM-based optimization. We plan to explore a lightweight variant of MASAM that retains its SOTA performance while reducing computational overhead.

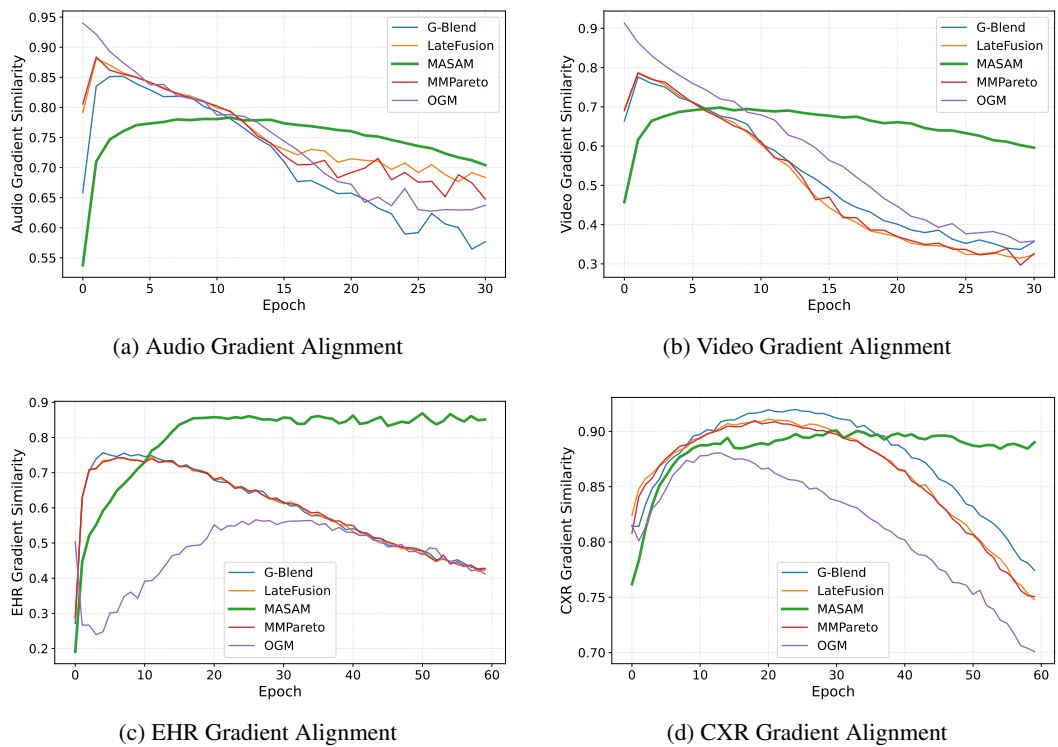

(a) Audio Gradient Alignment

(b) Video Gradient Alignment

(c) EHR Gradient Alignment

(d) CXR Gradient Alignment

Figure 15: Gradient alignment curves for the audio and video modalities in KS and the EHR and CXR modalities in MIMIC-Phenotype, quantified by the cosine similarity between each unimodal gradient and the fused objective gradient.

Table 10: Computational efficiency comparison across methods. Average Training Latency is measured in milliseconds, and Memory reports the peak allocated GPU memory during inference.

| Method | Average Inference Time (ms) | Average Training Time (ms) | Memory (MB) |
| --- | --- | --- | --- |
| MMPareto | 8.48 | 120.92 | 2482 |
| DrFuse | 19.77 | 19.99 | 3691 |
| AGM | 13.14 | 30.96 | 2027 |
| OGMGE | 8.80 | 20.95 | 2042 |
| MLA | 9.10 | 41.59 | 1921 |
| MedFuse | 13.93 | 20.35 | 1874 |
| Late Fusion | 8.64 | 19.09 | 1933 |
| AUG | 16.93 | 48.11 | 1926 |
| InfoReg | 12.97 | 140.64 | 2152 |
| **MASAM** | **8.88** | **186.63** | **2373** |

