# OpenReview forum: "MASAM: Multimodal Adaptive Sharpness-Aware Minimization for Heterogeneous Data Fusion"
_ICLR.cc/2026/Conference — ICLR 2026 Poster_

### Official Review · Reviewer_A3G6 · 2025-10-30

**Soundness:** 3
**Presentation:** 3
**Contribution:** 2
**Rating:** 4
**Confidence:** 4

**Summary:**

This paper introduces MASAM, a multimodal optimization framework that adapts Sharpness-Aware Minimization (SAM) to address modality imbalance in multimodal learning. The authors identify two main issues when applying SAM to multimodal settings—amplified modality imbalance and modality-agnostic perturbation—and propose two components: the Adaptive Perturbation Score (APS) and Modality-Decoupled Perturbation Scaling (MDPS). APS dynamically identifies dominant modalities based on convergence speed and gradient alignment, while MDPS decouples perturbations to reduce inter-modal interference. The method demonstrates improved performance across multiple multimodal datasets, achieving flatter minima and better balance among modalities.

**Strengths:**

1. The paper presents an interesting and well-motivated attempt to extend SAM to multimodal learning, where convergence dynamics differ across modalities.
2. The introduction of APS and MDPS is intuitive and addresses the challenge of applying sharpness-aware optimization in multimodal settings.
3. The experimental results are extensive and consistent, showing that MASAM achieves improved generalization and balanced performance across diverse datasets.
4. The visualizations are impressive.

**Weaknesses:**

1. Lines 90–92 state that “SAM’s uniform sharpness minimization thus disproportionately benefits dominant modalities with faster convergence.” However, in the proposed MASAM framework, SAM is applied only to dominant modality-specific encoders. This design choice appears contradictory to the stated motivation—if SAM inherently benefits dominant modalities, applying it exclusively to them seems counterintuitive. The logic connecting the problem statement and the solution needs clearer justification.
2. The introduction of unimodal auxiliary objectives and the APS mechanism represents another gradient-based modulation approach, which has been explored in existing works such as MMPareto, OGM, and related gradient alignment strategies. The paper would benefit from a clearer differentiation from these prior methods, especially in how APS or MDPS uniquely contributes beyond existing gradient-based optimization schemes.
3. In the discussion section (Line 422), the authors claim that “MASAM enables all modalities to converge jointly to flatter minima.” However, if a modality remains consistently weak (non-dominant) throughout training, no perturbation is applied to it. This raises doubts about whether MASAM truly ensures joint convergence across all modalities or only benefits dominant ones. A more detailed discussion of this limitation is warranted.

**Questions:**

1. The SAM technique seems to require multiple passes of gradient computation, which raises concerns about the efficiency of the proposed method.

I would consider raise my rating if the above concerns could be sufficiently addressed.

---

> ### Author Response · Authors · 2025-11-21
> **Responses to Weaknesses**
>
> Thank you very much for your careful and insightful review. We sincerely appreciate your thoughtful comments, which provide meaningful guidance and greatly help us improve the clarity and quality of our work.
>
> ---
>
> **Weakness 1： It appears contradictory that SAM reinforces the dominant modality, and the proposed method applies SAM only to the dominant modality. Further explain why this design choice is reasonable.**
>
> Thank you for the comment. We would like to respectfully clarify that they are not contradictory.
>
> First, applying SAM *uniformly to both modalities*, would reinforce the dominant modality. This is because the standard SAM, when uniformly applied to both modalities, would compute a single perturbation vector derived from the joint gradient. Since the dominant modality exhibits larger gradient magnitudes and faster convergence, it also dominates the direction of this shared perturbation. Consequently, applying this perturbation uniformly would **force the weaker modality to adapt to the dominant modality's optimization trajectory** rather than exploring its own loss landscape. This distorts the gradients for the weaker modality, exacerbating the imbalance.
>
> In contrast, MASAM dynamically identifies the dominant modality via the Adaptive Perturbation Score (APS). SAM is then exclusively applied to the dominant modality as a geometric regularization. Our analysis shows that dominant modalities tend to converge too quickly into sharp, unstable regions due to cross-modal interference. By applying SAM exclusively to the dominant modality, we force it to converge to a flat minimum while avoiding interfering with the optimization of the weaker modality. Meanwhile, we design the Modality-Decoupled Perturbation Scaling (MDPS), which ensures the dominant modality is regularized without generating the coupled interference that suppresses the weaker modality.
>
> We will revise the paper to better clarify this rationale.
> ___
>
>
>
> **Weakness 2: A clearer differentiation between the proposed APS or MDPS and existing gradient-based modulation approaches.**
>
> Thank you for the constructive comment. Existing methods like OGM, AGM, and G-Blend primarily focus on *gradient magnitude modulation*. They address modality imbalance by dynamically rescaling the magnitude of gradients to synchronize convergence speeds across modalities. However, these approaches largely overlook the **geometric properties** of the loss landscape (i.e., sharpness). Consequently, even if they balance the training speed, the model may still converge to sharp, unstable minima that generalize poorly.
>
> In contrast, MASAM focuses on geometric regularization by applying SAM on the dominant modality to explicitly minimize the sharpness of the loss landscape. APS (Adaptive Perturbation Score) measures both the learning speed and gradient alignment between the unimodal loss and the fusion loss to identify the dominant modality. MDPS (Modality-Decoupled Perturbation Scaling) scales the *perturbation vector* used by SAM, instead of the gradient magnitude. This ensures that the search for flat minima is aligned with the shared fusion objective, effectively decoupling the geometric regularization from the cross-modal interference.
>
> We will include relevant discussions in the revised paper.
> ___
>
> **Weakness 3: If a modality is consistently weak throughout training, no perturbation is applied to it, raising doubts whether MASAM ensures joint convergence.**
>
> Thank you for the insightful comment. We would like to clarify that the selection mechanism of MASAM ensures that any modality exhibiting sharp convergence behaviours will be selected to apply the SAM perturbation. Specifically, if a modality is in a sharp region or converging too rapidly, it will exhibit a large loss decay, resulting in a high APS score. Then, it will be identified as the dominant modality and will be subsequently selected to apply the SAM perturbation. Conversely, if a modality is not selected throughout training, it implies that its loss decay is small and its training trajectory is stable, further indicating that this modality is already in a relatively flat region and does not require additional perturbation.
>
> We will include relevant discussions in the revised paper.

---

> ### Author Response · Authors · 2025-11-21
> **Response to the Question**
>
> **Question: The SAM technique seems to require multiple passes of gradient computation, which raises concerns about the efficiency of the proposed method.**
>
> Thank you for your insightful comments regarding the computational efficiency. MASAM indeed requires multiple passes of gradient computation during training, leading to moderate training overhead. However, for inference, the proposed MASAM has the same complexity and running time as all baseline methods with a late fusion architecture.
>
> We report the per-step training time and peak GPU memory usage of all baselines on the MIMIC-Phenotype task. MASAM requires more GPU memory and training time than SAM due to separate gradient computations for unimodal and fusion losses. MMPareto has similar computational requirements but higher memory usage than MASAM. AGM, MLA, and OGM use joint learning with comparable memory usage, though MLA has longer training time from alternating modality updates. MedFuse reduces costs by using RNN-based sequential fusion instead of feature concatenation. DrFuse has the highest memory usage due to its complex architecture with separate encoders for distinct and shared representations. While MASAM introduces moderate training overhead, its significant performance gains justify this tradeoff in many multimodal scenarios.
>
> _Table R12: Computational efficiency comparision_
>
> | Method      | Per-step Time (ms) | Memory (MB) |
> |-------------|---------------------|-------------|
> | MASAM       | 123.24              | 2372.87     |
> | MMPareto    | 75.19               | 2481.68     |
> | SAM         | 62.18               | 2062.13     |
> | DrFuse      | 19.99               | 3690.51     |
> | AGM         | 10.22               | 2026.85     |
> | OGMGE       | 10.72               | 2042.10     |
> | MLA         | 22.76               | 1921.28     |
> | MedFuse     | 8.93                | 1874.35     |
> | Late Fusion | 8.59                | 1932.75     |
>
> ---
>
> Again, we appreciate your constructive and insightful comments.

---

> ### Comment · Reviewer_A3G6 · 2025-11-25
>
> I thank the authors for the detailed response. Their explanation resolves most of my concerns, and I have raised my rating to 6. With that being said, I have a few recommendations:
>
> Firstly, I suggest that the authors refine the abstract and introduction to explicitly state that MASAM prevents weaker modalities from being affected by misaligned perturbations from the dominant modality. This is a more precise and compelling motivation than simply stating that the calculation is "affected by interference" or "exacerbating modality imbalance."
>
> Secondly, while inference is efficient, the significant training overhead (approx. 15x that of Late Fusion in Table R12) should be transparently discussed as a limitation, which is currently missing.
>
> Finally, it would be beneficial if the authors could clarify how the "dominant vs. weak" selection mechanism extends to scenarios with three or more modalities.
>
> Thank you again for this solid work.

---

> > ### Author Response · Authors · 2025-11-25
> >
> > Thank you for your thoughtful follow-up and for your efforts in helping us improve the work. We sincerely appreciate your suggestions. To sum up, we will incorporate these improvements into the final version of the paper:  1) we will revise the abstract and introduction with a clearer and more precise motivation; 2) we will discuss the training-time overhead as a limitation; 3) we will clarify how the selection mechanism generalizes to three or more modalities.
> >
> > Thank you again for your time!

---

### Official Review · Reviewer_EFgG · 2025-11-01

**Soundness:** 3
**Presentation:** 4
**Contribution:** 3
**Rating:** 8
**Confidence:** 3

**Summary:**

In this paper, the authors presented MASAM, a novel method that applies sharpness-aware minimization (SAM) to multimodal setting. The paper identifies the issue with directly applying SAM to multimodal setting where modality imbalance could be exacerbated by SAM, and devised a method to decouple the modalities when determining the scales of the perturbations. The proposed method, MASAM, first determines the dominant modality using APS, then applies perturbation on the dominant modality using MDPS. The proposed method was evaluated over 6 tasks across 5 very different datasets from different domains against a large set of baseline fusion methods, and MASAM achieves top performance across all tasks. Additional analysis was conducted to highlight MASAM's robustness to label noise and ability to allow all modalities to converge jointly to a flatter minima, and ablation studies was conducted to demonstrate the necessity for both APS and MDPS modules.

**Strengths:**

1. The proposed method outperforms many baselines over 5 very different domains, demonstrating MASAM's generalizability across many multimodal application domains.

2. The overall presentation quality of the paper is good. It is overall easy to follow.

3. There is theoretical analysis to show that naive SAM does not work well under multimodal settings, and that MASAM can better converge all modalities.

4. There is additional visualizations and empirical analysis to demonstrate MASAM's more balanced learning across modalities and allows all modalities to converge to a flatter minimum. The additional analysis also shows MASAM's better robustness to label noise.

5. There is ablation study to demonstrate the necessity of both APS and MDPS modules.

**Weaknesses:**

The paper could be improved by including more justification/analysis for the necessity of both Decay and $\gamma$ in APS. In figure 10a, it seems like low $\alpha$ generally performs quite well, so is Decay actually necessary? There needs to be an ablation study to justify this (i.e. what happens if you set $\alpha$ to 0?)

**Questions:**

I am a bit confused about the setup for calculating Decay. Can you explain why it is calculated as the difference between the previous round's absolute loss and the current round's moving average? Also, in Figure 10b, it seems like a high $\beta$ works better, but wouldn't a higher $\beta$ make the Decay smaller and more likely to be zero?

---

> ### Author Response · Authors · 2025-11-21
> **Thank you for your constructive comments.**
>
> Thank you very much for your thoughtful and constructive review. We truly appreciate your insights, and we hope that our responses and additional analyses address your concerns satisfactorily.
> ___
> **Weakness 1: The paper lacks sufficient justification for the necessity of both the Decay term and the coefficient $\gamma$ in APS, and requires an ablation study to validate the effect of removing the Decay component (i.e., setting $\alpha=0$).**
>
> Decay and $\gamma$ are both necessary because they capture two distinct but complementary dimensions of modality dominance. Decay measures the learning speed for each modality, if a modality is having a significantly faster speed of loss decrease, it indicates that the modality is in a relatively sharp region. $\gamma$ measures the alignment between unimodal gradient and the fusion gradient, it identifies modalities that are dominating the direction of the joint optimization.
>
> As suggested, we include a further ablation study by setting $\alpha=0$. We summarize the results in the following table.
>
> *Table R10: Comparison of Alternative APS Designs in MIMIC-Phenotype and Kinetics-Sounds(KS)*
>
> | Method                    | KS Accuracy (± std) | $\Delta$ Acc (%) | MIMIC-Phenotype AUPRC (± std) | $\Delta$ AUPRC (%) |
> |---------------------------|----------------------|-----------|-------------------------------|--------------|
> | MASAM (Default)           | **0.740 ± 0.0084**   | 0.00%     | **0.498 ± 0.0010**            | 0.00%        |
> | MASAM with $\alpha$ = 0   | 0.733 ± 0.0043       | -0.95%    | 0.489 ± 0.0033                | -1.81%       |
>
>
> Results show that if the Decay component is removed from APS, the performance drops 0.95% and 1.81% for KS and MIMIC-Phenotype datasets, respectively. This validates that Decay is a necessary component in the APS module.
>
>
> ___
>
> **Q1: Why decay is calculated as the difference between the previous round's absolute loss and the current round's moving average?**
>
> The Decay term is calculated as $\text{Decay}_m^{(t)} = \max \left( 0, \mathcal{L}_m^{(t-1)} - \text{MA}_m^{(t)} \right)$, which is designed to measure the convergence speed. The previous loss $\mathcal{L}_m^{(t-1)}$ serves as a reference point while the moving average $\text{MA}_m^{(t)}$ represents a smoothed history of the loss. Their differences quantify the magnitude of the recent loss reduction, thus reflecting the sharpness for the modality and if the modality is a dominate one.
>
>
> ___
>
> **Q2: A higher $\beta$ appears to yield better results, but wouldn't a larger $\beta$ cause the decay to become smaller and more readily approach zero?**
>
> A larger momentum value (e.g., $\beta=0.9$) would not cause the decay to vanish, instead, it ensures the stability of calculating the decay. If $\beta$ is small, the moving average $\text{MA}_m^{(t)}$ updates rapidly and tracks the instantaneous loss $\mathcal{L}_m^{(t)}$ very closely, then the Decay term would capture only mini-batch noise rather than meaningful learning progress. Therefore, a higher $\beta$ value (e.g., 0.9) ensures that $\text{MA}_m^{(t)}$ serves as a stable, long-term baseline.
>
> We further conducted sensitivity analysis for $\beta$ on the KS dataset. The results are summarized as below. They show that $\beta=0.9$ is a proper configuration, which yields the best performance.
>
> *Table R11: Sensitivity analysis for $\beta$, ★ indicates the best performance*
>
> | $\beta$ | Accuracy              | $\Delta$ Accuracy (%) |
> |--------|------------------------|------------------------|
> | 0.1    | 0.730±0.0037           | -1.35%                 |
> | 0.2    | 0.732±0.0061           | -1.08%                 |
> | 0.3    | 0.731±0.0111           | -1.22%                 |
> | 0.5    | 0.728±0.0042           | -1.62%                 |
> | 0.7    | 0.731±0.0061           | -1.22%                 |
> | 0.9    | **0.740±0.0047 ★**     | 0.00%                  |
> | 0.99   | 0.736±0.0037           | -0.54%                 |

---

> > ### Comment · Reviewer_EFgG · 2025-11-23
> >
> > Thank you for your response! My score remains positive.

---

> > > ### Author Response · Authors · 2025-11-25
> > >
> > > Thank you for your timely follow-up and positive assessment of our work. We truly appreciate your time and constructive comments.

---

### Official Review · Reviewer_TTzL · 2025-11-01

**Soundness:** 3
**Presentation:** 3
**Contribution:** 3
**Rating:** 6
**Confidence:** 4

**Summary:**

This paper presents MASAM (Multimodal Adaptive Sharpness-Aware Minimization) to address the challenges of modality imbalance and heterogeneous loss geometries in multimodal learning. The authors introduce Adaptive Perturbation Score (APS) and Modality-Decoupled Perturbation Scaling (MDPS) to ensure balanced multimodal learning while benefiting from Sharpness-Aware Minimization (SAM). Their approach is demonstrated to achieve flatter loss landscapes for each modality, resulting in improved generalization across diverse datasets and tasks.

**Strengths:**

- The paper presents both qualitative and quantitative experiments across five multimodal datasets, with six downstream tasks, demonstrating MASAM's effectiveness in diverse domains, including clinical, video, and image-text datasets.
- The problem of modality imbalance is well-motivated, and the proposed solution is thoroughly explained. The use of SAM in multimodal learning is a novel approach to addressing this problem.
- The writing is clear, logical, and easy to follow.

**Weaknesses:**

- The loss landscape analysis is only conducted on the small clinical MIMIC dataset. It would be more convincing if similar visualizations were provided for larger and more diverse "in-the-wild" datasets, ensuring the scalability and generalizability of the approach across various real-world data settings.
- The sensitivity of hyperparameters, such as $\alpha$ and $\rho$, is not sufficiently explored. A more detailed analysis of how these parameters vary across different datasets and how they affect the method performance is necessary.
- The paper focuses on experiments with two modalities (e.g., EHR + CXR, audio-video). An exploration of how MASAM performs with more than two modalities would be valuable for assessing its scalability in more complex multimodal fusion tasks.

Minor Issues:

The used "Kinetics" dataset should be called the "Kinetics-Sounds" dataset. Kinetics-Sounds dataset is a subset of Kinetics.

**Questions:**

Please check the above section.

---

> ### Author Response · Authors · 2025-11-21
> **Responses to Weaknesses 1 and 2**
>
> Thank you very much for your constructive and insightful feedback. Your comments are highly valuable and have helped us further improve the clarity and rigor of our work. We hope that the following responses satisfactorily address your concerns.
>
> ---
>
> **Weakness1: Loss landscape analysis is only conducted on MIMIC; it would be more convincing if visualizations were also provided for other datasets.**
>
> As suggested, we have conducted loss landscape analysis on all other datasets with visualizations updated in the revised version (see Appendix E.2): CREMA-D (Fig. 10), Kinetics-Sound (Fig. 11), Food101 (Fig. 12), and ADNI (Fig. 13). For all datasets, we observe the similar patterns as in Fig. 5 of our original paper: MASAM consistently achieves flat solutions compared with existing methods.
>
> ---
>
>
> **Weakness 2: The sensitivity of key hyperparameters (e.g., $\alpha$ and $\beta$) is not sufficiently explored. It needs to explore across different datasets.**
>
> As suggested, we also conduct sensitivity analysis on Kinetics-Sounds and MIMIC datasets for the following key hyperparameters:
> - $\alpha$: the APS parameter, controlling the balance between learning speed and gradient consistency in determining the dominance of each modality.
> - $\beta$: the momentum parameter, controlling the smoothness of the moving average.
> - $\rho$: the perturbation radius, controlling the step size of the perturbation.
>
> Results can be found in Fig. 14 of the revised paper. For easier reference, we summarize the results for Kinetics-Sounds in tables below, where ★ indicates the best performance.
>
> *Table R6: Sensitivity analysis for $\alpha$*
>
> | $\alpha$ | Accuracy              | $\Delta$ Accuracy (%) |
> |---------|------------------------|------------------------|
> | 0.1     | 0.729±0.0044           | -1.49%                 |
> | 0.2     | 0.737±0.0042           | -0.41%                 |
> | 0.3     | **0.740±0.0047 ★**     | 0.00%                  |
> | 0.5     | 0.732±0.0061           | -1.08%                 |
> | 0.7     | 0.731±0.0077           | -1.22%                 |
> | 0.9     | 0.724±0.0042           | -2.16%                 |
>
>
> *Table R7: Sensitivity analysis for $\beta$*
>
> | $\beta$ | Accuracy              | $\Delta$ Accuracy (%) |
> |--------|------------------------|------------------------|
> | 0.1    | 0.730±0.0037           | -1.35%                 |
> | 0.2    | 0.732±0.0061           | -1.08%                 |
> | 0.3    | 0.731±0.0111           | -1.22%                 |
> | 0.5    | 0.728±0.0042           | -1.62%                 |
> | 0.7    | 0.731±0.0061           | -1.22%                 |
> | 0.9    | **0.740±0.0047 ★**     | 0.00%                  |
> | 0.99   | 0.736±0.0037           | -0.54%                 |
>
>
> *Table R8: Sensitivity analysis for $\rho$*
> | $\rho$ | Accuracy              | $\Delta$ Accuracy (%) |
> |-------|------------------------|------------------------|
> | 0.1   | 0.680±0.0062           | -8.11%                 |
> | 0.2   | 0.709±0.0027           | -4.19%                 |
> | 0.3   | 0.726±0.0065           | -1.89%                 |
> | 0.5   | 0.737±0.0056           | -0.41%                 |
> | 0.7   | **0.740±0.0047 ★**     | 0.00%                  |
> | 0.9   | 0.738±0.0018           | -0.27%                 |
>
>
> Across all three hyperparameters, MASAM shows strong robustness.
>
> For APS parameter **$\alpha$**, which serves as the coefficient that jointly controls loss-decay reweighting and gradient-alignment strength, the model consistently outperforms the baseline, achieving peak performance around $0.2$ to $0.3$. This indicates that properly balancing these two components is crucial for achieving effective gradient alignment on this dataset.
>
> For the momentum parameter $\beta$, MASAM again maintains stable accuracy across all tested values, confirming the benefit of incorporating long-term loss trends.
>
> For the perturbation radius $\rho$, although the accuracy varies more noticeably in the small $\rho$ region (0.1–0.3), the performance becomes stable and consistently strong for $\rho \ge 0.3$, reaching its maximum at larger perturbation radii (0.7–0.9). These results also show **a dataset-dependent pattern**: whereas MIMIC-Phenotype attains its best performance around $\rho=0.4$, Kinetics-Sounds favors a larger range ($\rho=0.7$). This difference arises because datasets exhibit distinct loss landscape geometries, leading to different optimal choices of $\rho$.

---

> ### Author Response · Authors · 2025-11-21
> **Response to Weakness 3**
>
> **Weakness 3: The paper only studies two modalities; evaluating MASAM on three-modality datasets would better show its scalability.**
>
> As suggested, we include an additional dataset, **UR-Funny** [1], which contains three modalities (text, video, and audio) with labels for humor detection. We follow the data processing procedures described in AGM [2]. Due to time and resource constraints, we adopt late fusion and AGM as baselines, both of which natively offer three-modality implementation on this task. Results are summarized in the table below, which shows the scalability of MASAM to three modalities. Specifically, MASAM ahieves relative performance gain of 3.39% and 1.75% against Late Fusion and AGM, respectively.
>
> *Table R9: Additional experimental results of humor detection using three modalities on the UR-Funny dataset.*
> | Method      | Accuracy         |
> | ----------- | ---------------- |
> | Late Fusion | 0.620±0.0164     |
> | AGM         | 0.630±0.0091     |
> | **MASAM**   | **0.641±0.0058** |
>
> [1] MK Hasan, et al. UR-FUNNY: A multimodal language dataset for understanding humor. *EMNLP*. 2019.
> [2] Li H, et al. Boosting multi-modal model performance with adaptive gradient modulation. *ICCV*. 2023.
>
> ___
>
> **Minor Issue: "Kinetics" dataset should be called the "Kinetics-Sounds" dataset.**
>
> We have replaced "Kinetics" with "Kinetics-Sounds" in the revised paper.
>
> ---
> We thank the reviewer again for constructive comments and suggestions.

---

> > ### Comment · Reviewer_TTzL · 2025-11-26
> >
> > Thanks for the author's response. My concerns are well addressed. And I will keep my positive score.

---

### Official Review · Reviewer_U8Y1 · 2025-11-01

**Soundness:** 3
**Presentation:** 2
**Contribution:** 2
**Rating:** 4
**Confidence:** 4

**Summary:**

This paper proposes the MASAM method, which introduces the Sharpness Aware Minimization (SAM) strategy from machine learning into multimodal learning. By adjusting the loss function to prioritize model robustness, it enables each modality to
explore flatter minima along directions that align with shared information while remaining robust to heterogeneous loss landscapes, thereby improving overall performance.

**Strengths:**

⦁	This paper introduces Sharpness-Aware Minimization into the multimodal domain and further designs it specifically for the multimodal context, enhancing its applicability.

**Weaknesses:**

⦁	The baselines are weak. Only 1 work published in 2024 is compared and no works published in 2025 are compared. This comparision is not enough to show its SOTA performance.
⦁	 The writing is confusing. Fig 1b is not mentioned in the manuscript. Fig 2 is used until in page 6.
⦁	Tha ablation study should be performed on more datasets. Also, APS and MDPS should be compared to other choices to verift their advantages.
⦁	The experimental section primarily focuses on interpreting dataset performance while neglecting an in-depth analysis of the method’s effectiveness.
⦁	The authors argue that MDPS aligns gradients between unimodal and fusion objectives, please give detailed verification.

**Questions:**

Refer to the weaknesses.

---

> ### Author Response · Authors · 2025-11-21
> **Response to Weaknesses 1 and 2**
>
> We thank the reviewr for the constructive and detailed suggestions, which help us greatly improve the clarity and quality of the paper.
> ___
>
> **Weakness 1: Limited comparisons with works published in 2025.**
>
> Thank you for the constructive comment. As suggested, we further include two recent baselines, InfoReg (CVPR-25) [1] and AUG (NeurIPS-25) [2]. Results across datasets are summarized in the table below. As shown, MASAM consistently outperforms the latest methods in this area. This part has been be incorporated into Section 4 in our revised paper.
>
> *Table R1: Results of Additional Recent Baseline Models.*
> | Method        | ADNI (mAP)        | KS (Accuracy)     | CREMAD (Accuracy)  | Food101 (Accuracy) | MIMIC Phenotype (AUPRC) | MIMIC Mortality (AUPRC) |
> |--------------|--------------------|--------------------|---------------------|---------------------|---------------------------|---------------------------|
> | InfoREG [1]      | 0.820±0.0135       | 0.671±0.0103       | 0.723±0.0054        | 0.925±0.0008        | 0.481±0.0017              | 0.581±0.0105              |
> | AUG [2]          | 0.847±0.0068       | 0.689±0.0083       | 0.770±0.0152        | 0.923±0.0015                   | 0.472±0.0031                         | 0.579±0.0120                         |
> | **MASAM (ours)** | **0.857±0.0042** | **0.740±0.0084**   | **0.814±0.0046**    | **0.935±0.0011**    | **0.498±0.0010**          | **0.603±0.0086**          |
>
>
> [1] C. Huang, Y Wei, Z. Yang, D. Hu. Adaptive unimodal regulation for balanced multimodal information acquisition. *CVPR*. 2025.
> [2] Q. Jiang, L. Huang, Y. Yang. Rethinking multimodal learning from the perspective of mitigating classification ability disproportion. *NeurIPS*. 2025.
>
>
> ___
>
> **Weakness 2: Fig. 1b is not mentioned and Fig. 2 is not referenced until Page 6.**
>
> Thank you for the comments, we have revised the introduction section to include proper references to both Fig. 1b and Fig. 2.

---

> > ### Comment · Reviewer_U8Y1 · 2025-11-26
> >
> > Thanks for providing these additional experiments. For Table R1, please clarify the comparison settings. Whether these results are reported in their papers or implemented with the same setting in the proposed work.

---

> > > ### Author Response · Authors · 2025-11-26
> > >
> > > Thank you for the follow-up question. The additional methods, InfoREG and AUG, are both implemented with the same settings in the proposed work to ensure fair comparisons. We will add this clarification to our revised paper.

---

> > > > ### Comment · Reviewer_U8Y1 · 2025-11-28
> > > >
> > > > Thanks. This further addresses my concerns.

---

> ### Author Response · Authors · 2025-11-21
> **Response to Weakness 3**
>
> **Weakness 3: The ablation study should be extended to additional datasets, and APS/MDPS should be compared with more alternatives to verify their advantages.**
>
>
> Thank you for the suggestion. We have (1) conducted ablation studies on additional dataset Kinetics-Sounds, and (2) compared alternatives to APS/MDPS on both MIMIC-Phenotype and Kinetics-Sounds. These two part have been incorporated into Section 4.4 and Section E.6, respectively, in our revised paper.
>
> *Table R2: Results of Ablation Studies on the Kinetics-Sounds Dataset.*
> | Kinetics-Sounds     | Method      | APS | MDPS | SAM | Accuracy  | vs. w/o APS | vs. w/o MDPS | vs. SAM Only | vs. Late Fusion |
> | ----- | ----------- | --- | ---- | --- | --------- | ----------- | ------------ | ------------ | --------------- |
> | **–** | **MASAM**   | ✔   | ✔    | ✔   | **0.740** | +2.21%      | +2.35%       | +7.40%       | +16.35%         |
> | 1     | w/o APS     | ✘   | ✔    | ✔   | 0.724     | —           | +0.14%       | +5.08%       | +13.84%         |
> | 2     | w/o MDPS    | ✔   | ✘    | ✔   | 0.723     | -0.14%      | —            | +4.93%       | +13.68%         |
> | 3     | SAM Only    | ✘   | ✘    | ✔   | 0.689     | -4.84%      | -4.70%       | —            | +8.33%          |
> | 4     | Late Fusion | ✘   | ✘    | ✘   | 0.636     | -12.11%     | -12.06%      | -7.62%       | —               |
>
> On the Kinetics-Sounds dataset, the ablation study shows that both APS and MDPS contribute clear and complementary gains. Removing APS reduces accuracy by 2.21%, and comparing w/o-MDPS with SAM-Only shows APS provides an additional +4.93% improvement. Similarly, MDPS brings +2.35% over w/o-MDPS and +5.05% over SAM-Only. Directly applying SAM is ineffective in this multimodal setting (–7.47% vs. MASAM). With APS and MDPS jointly enabled, MASAM achieves the best performance (0.740, +16.35% over Late Fusion), confirming that both components are necessary for stable and balanced optimization on audio–video data. We have updated the result in Table 3 in the revised paper.
>
>
> To further verify the importance of our design choices, we also evaluate several alternatives for APS and MDPS.
>
>
> (i) For **APS**, we adopt different scoring strategies:
> - Using the scoring scheme from the OGM paper.
> - Setting the parameter $\alpha$ to zero, which removes the loss decay term and keeps only $\gamma_m^{(t)}$.
>
> (ii) For **MDPS**, we experiment with different gradient choices:
> - Using the difference between the fusion and unimodal gradients:
> $\mathbf{g}_ 1^\prime=\nabla_ {\boldsymbol{\theta}m} \mathcal{L}_ {fuse} - \nabla_ {\boldsymbol{\theta}_ m} \mathcal{L}_ m$.
> - Using the component of $\nabla_ {\boldsymbol{\theta}_ m} \mathcal{L}_ m$ that is orthogonal to the fusion gradient
> $\mathbf{g}_ 2^\prime=\nabla_ {\boldsymbol{\theta}_ m} \mathcal{L}_ {fuse}$.
>
>
> The results on MIMIC and KS datasets are as below. The results demonstrate the superiority of APS and MDPS against their potential alternatives.
>
>
>
> *Table R3: Comparison of Alternative APS and MDPS Designs in MIMIC-Phenotype and Kinetics-Sounds(KS)*
>
> | #  | Method                    | KS Accuracy (± std) | $\Delta$ Acc (%) | MIMIC-Phenotype AUPRC (± std) | $\Delta$ AUPRC (%) |
> |----|---------------------------|----------------------|-----------|-------------------------------|--------------|
> | –  | MASAM (Default)           | **0.740 ± 0.0084**   | 0.00%     | **0.498 ± 0.0010**            | 0.00%        |
> | 1  | MASAM w/ OGM Score        | 0.731 ± 0.0012       | -1.22%    | 0.493 ± 0.0013                | -1.00%       |
> | 2  | MASAM w/ APS ($\alpha$ = 0)      | 0.733 ± 0.0043       | -0.95%    | 0.489 ± 0.0033                | -1.81%       |
> | 3  | MASAM w/ $\mathbf{g}_1^\prime$              | 0.700 ± 0.0064       | -5.41%    | 0.462 ± 0.0026                | -7.23%       |
> | 4  | MASAM w/ $\mathbf{g}_2^\prime$              | 0.721 ± 0.0074       | -2.57%    | 0.489 ± 0.0010                | -1.81%       |
>
> We first examine the APS variants (\#1 and \#2). Using the OGM scoring strategy (\# 1) reduces the KS accuracy by 1.22% and the MIMIC-Phenotype AUPRC by 1.01%. Removing the loss-decay term(\# 2) also decreases performance by 0.95% on KS and 1.81% on MIMIC-Phenotype. These results indicate that the APS alternatives fail to reliably identify the correct dominant modality for SAM optimization.
>
> Next, we evaluate the MDPS variants (\#3 and \#4). The gradient-difference variant $\mathbf{g}_1^\prime$ (\# 3) causes substantial degradation (5.41% on KS and 7.23% on MIMIC-Phenotype). The orthogonal-gradient variant $\mathbf{g}_2^\prime$ (\#4) also performs worse than MASAM on both datasets, with drops of 2.57% on KS and 1.81% on MIMIC-Phenotype. These results show that the MDPS alternatives cannot effectively remove cross-modal interference during SAM perturbation. The content is updated in Table 9 in the revised paper.

---

> ### Author Response · Authors · 2025-11-21
> **Responses to Weaknesses 4 to 5**
>
> **Weakness 4: The experimental section primarily focuses on interpreting dataset performance while neglecting an in-depth analysis of the method’s effectiveness**
>
>
> Thank you for the comments. Other than quantitative evaluations, we also perform (1) loss-landscape visualization, (2) unimodal performance evaluation, and (3) theoretical analysis to analyze the effectiveness of the proposed model from different aspects.
>
> (1) **Loss-landscape analysis:** Fig. 5 and Appendix E.2 (added in the revision) visualize the loss landscapes for all datasets, showing that MASAM consistently leads to flatter and more stable minima compared with existing methods. This benefits from i) **APS**, which selects the dominant modality and prevents the weak modality from being incorrectly perturbed; and ii) **MDPS**, which mitigates cross-modal interference when applying SAM perturbation to the selected strong modality.
>
>
> (2) **Unimodal evaluation:** We examine the unimodal encoder quality by freezing the encoder after multimodal training, with a classifier head on top. Fig. 4 shows that MASAM achieves consistently superior unimodal accuracy, indicating more balanced multimodal learning.
>
>
> (3) **Theoretical analysis:** Theorem 1 (proof in Appendix D) establishes convergence of MASAM to a stationary point of the objective in Eq. (7), providing additional support for method soundness.
>
>
> These analyses are discussed in Section 4.2 *Modality Balance and Loss Landscape Flatness* and 3.2 *Theoretical Analysis* in the revised paper, collectively demonstrating the effectiveness of MASAM.
>
>
>
> ---
>
> **Weakness 5: The authors argue that **MDPS aligns** gradients between unimodal and fusion objectives, please give detailed verification.**
>
> To further verify that MDPS achieves gradient alignment between unimodal and fusion objectives, we conduct additional experiments to directly measure the gradient alignment by computing the cosine similarity between the gradients of the unimodal and fusion objectives on Kinetics-Sounds dataset during training. The results are visualized in Fig. 15 of the revised paper. Due to format restrictions, we summarize the average cosine similarities in different training epochs in tables below for easier references. We have provided gradient alignment figures in Section E.7 of our revised paper.
>
> *Table R4: Gradient Alignment for the Audio Modality.*
>
> | Method         | Epochs 0–5       | Epochs 5–11  | Epochs 11–17 | Epochs 17–23 | Epochs 23–29 |
> | -------------- | --------- | ----- | ----- | ----- | ----- |
> | **MASAM**      | 0.716     | 0.778 | **0.777** | **0.760** | **0.733** |
> | **GBlend**     | 0.811     | 0.813 | 0.736 | 0.657 | 0.606 |
> | **LateFusion** | 0.849     | **0.821** | 0.753 | 0.717 | 0.694 |
> | **MMPareto**   | 0.850     | **0.821** | 0.748 | 0.701 | 0.677 |
> | **OGM**        | **0.887** | 0.818 | 0.763 | 0.674 | 0.636 |
>
> *Table R5: Gradient Alignment for the Video Modality.*
>
> | Method         | Epochs 0–5       | Epochs 5–11  | Epochs 11–17 | Epochs 17–23 | Epochs 23–29 |
> | -------------- | --------- | ----- | ----- | ----- | ----- |
> | **MASAM**      | 0.632     | 0.693 | **0.683** | **0.660** | **0.628** | **0.599** |
> | **GBlend**     | 0.731     | 0.669 | 0.526 | 0.409 | 0.359 | 0.347 |
> | **LateFusion** | 0.742     | 0.662 | 0.495 | 0.372 | 0.331 | 0.319 |
> | **MMPareto**   | 0.742     | 0.662 | 0.501 | 0.378 | 0.336 | 0.312 |
> | **OGM**        | **0.825** | **0.717** | 0.603 | 0.461 | 0.384 | 0.356 |
>
>
>
> They demonstrate that MASAM achieves significantly greater alignment between the unimodal and fusion objectives compared to other baselines as the training going on.
>
> ---
> We thank the reviewer for the constructive comments. The results shown above has been updated in the revised paper.

---

> > ### Comment · Reviewer_U8Y1 · 2025-11-26
> >
> > The additional experiments address part of my concerns.

---

### Comment · Area_Chair_7XiX · 2025-11-24

Dear reviewers,

Thank you for your dedicated service as reviewers. Your efforts are critical to the success of our conference, and we deeply appreciate your time and expertise.

This paper has received reviews from reviewers but some have not provided a response to the author rebuttal. Given the limited time we have for author-reviewer discussions, we kindly ask you to share your post-rebuttal feedback to help clarify your perspective and aid the decision-making process.

Your input is invaluable in ensuring a fair and thorough review process.

Best,
AC

---

### Author Response · Authors · 2025-11-30
**Summary of Rebuttal Discussions**

We sincerely thank all reviewers for their thoughtful and constructive feedback. Their comments helped us strengthen the technical soundness of the proposed MASAM method and the clarity of the paper. The main strengths highlighted by the reviewers include:

- Well-motivated problem formulation on modality imbalance (`Reviewer TTzL and A3G6`).
- A thorough analysis and rationale for why heterogeneous convergence dynamics require special treatment (`Reviewer TTzL and A3G6`) and why naive SAM does not work in multimodal settings (`Reviewer EFgG`).
- Novel adaptation of SAM to multimodal learning via Adaptive Perturbation Score (APS) and Modality-Decoupled Perturbation Scaling (MDPS) (`All four reviewers`)
- Strong generalizability and promising performance across diverse datasets and tasks (`Reviewer TTzL, EFgG, and A3G6`)
- Impressive loss-landscape visualizations (`Reviewer A3G6`) and analyses showing strong evidence of the effectiveness of MASAM (`Reviewer EFgG`).
- Clear writing and strong overall presentation quality (`Reviewer TTzL and EFgG`)

**Our Core Contributions:** Multimodal learning often suffers from the modality-imbalance problem. We identify that geometric factors (e.g., sharpness) play important roles in balancing multimodal learning. Although Sharpness-Aware Minimization (SAM) is a principled approach to search for flat minima, our theoretical analyses show that it fails in the multimodal setting due to modality imbalance. In this work, we design the Adaptive Perturbation Score (APS) and Modality-Decoupled Perturbation Scaling (MDPS) to select the dominant modality and apply modality-decoupled perturbation over the dominant modality. Empirical evaluation shows that MASAM consistently and significantly outperforms all baselines across six datasets and seven tasks.

During the rebuttal, we have added the following content to address concerns from the reviewers:

- **Additional baselines**: We further included two recent methods, InfoReg(CVPR-25) and AUG(NeurIPS-25), as baselines. Results are added in *Section 4*. MASAM consistently outperforms these latest methods. (`Reviewer U8Y1`)
- **Demonstrated scalability to three modalities**: We further applied MASAM to the **three-modality** UR-FUNNY dataset to evaluate its scalability, and MASAM obtains the best performance among the compared methods. These results show that MASAM scales effectively to datasets with different numbers of modalities. (`Reviewer TTzL`)
- **Ablation studies over more datasets**: We expanded our ablation analysis by including an additional dataset, Kinetics-Sounds(_Section 4.4_), and evaluated APS/MDPS alternatives on both MIMIC-Phenotype and Kinetics-Sounds (_Appendix E.6_). The results further verify the effectiveness of our design and the clear advantage of APS and MDPS over their alternatives. (`Reviewer U8Y1 and EFgG`)
- **More comprehensive analysis on model effectiveness and loss landscape**: We supplemented the 3D visualisation of the loss landscape across all datasets in _Appendix E.2_, demonstrating that MASAM conclusively enables multimodal convergence towards a flat region. (`Reviewer TTzL and U8Y1`)
- **Added gradient-alignment verification**: We directly quantified cosine similarities between unimodal and fusion gradients (_Appendix E.7_). (`Reviewer U8Y1 and A3G6`)
- **Added hyperparameter sensitivity analyses**: We added sensitivity analysis on Kinetics-Sounds and performed cross-dataset comparisons. Results confirm the hyperparameter robustness of MASAM (`Reviewer TTzL and EFgG`)
- **Computational efficiency analysis**: We additionally report per-step training and inference times and GPU memory usage for all baselines on MIMIC-Phenotype (_Appendix E.8_). (`Reviewer A3G6`)

We also clarified reviewers' conceptual and methodological concerns:
- **Differentiation from gradient-magnitude modulation methods**: We clarified that, unlike prior gradient-modulation methods, MASAM introduces a novel geometry-aware approach by applying SAM to explicitly minimize sharpness (_Appendix B.2_). (`Reviewer A3G6`)
- **Rationale of applying SAM only to the dominant modality**: We clarified that our design is not contradictory to the initial observation of SAM reinforcing the dominant modality, but instead is beneficial in the multimodal setting. (_Section 3.1 and Appendix H.1_). (`Reviewer A3G6`)
- **Ensuring joint convergence when a modality appears consistently weak**:
We clarified that if a modality is not selected to be perturbed ("consistently weak"), then it is already in a relatively flat region, since otherwise it will have a large APS and thus be selected. (_Section 3.1 and Appendix H.2_). (`Reviewer A3G6`)

***All reviewers have acknowledged our rebuttal and paper revisions. All reviewers have also stated that their concerns are addressed.*** Once again, we sincerely appreciate the encouraging feedback and insightful comments from all reviewers and the ACs.

---

### Meta-Review · Area_Chair_2qRK · 2026-01-10

**Summary:**

Reviewers rate the paper 4, 6, 8, and 4 (average 5.5). Main weaknesses were initial baseline coverage, limited loss-landscape and sensitivity analyses beyond the small clinical dataset, scalability beyond two modalities, clarity links between the problem and applying SAM only to dominant modalities, and efficiency concerns.

**Reviewer Concerns:**

The authors added recent 2025 baselines and showed consistent gains, extended loss-landscape visualizations to all datasets, performed sensitivity studies for alpha, beta,  and rho, added ablations and comparisons against APS/MDPS alternatives, provided direct gradient-alignment measurements (cosine similarity) over training, included a three-modality UR-Funny experiment, clarified the rationale for perturbing the dominant modality, and reported training-time/memory overhead; figure references and dataset naming were fixed.

The issues raised by the reviewers are well addressed.

**Reviewer Scores:**

Given the author's detailed response to the reviewers' questions, the two reviewers (EFgG, TTzL) who initially gave positive scores (8,6) will maintain their ratings, while the other two reviewers (U8Y1, A3G6) who gave lower scores (4,4) both indicate they would like to raise their ratings to positive.

---

### Decision · Program_Chairs · 2026-01-26

Accept (Poster)